# ACTIVATIONREASONING: LOGICAL REASONING IN LATENT ACTIVATION SPACES

**Lukas Helff**[1,2][*], **Ruben Härle**[1,3,4][*], **Wolfgang Stammer**[5][†] **Felix Friedrich**[6],
**Manuel Brack**[2,7] , **Antonia Wüst**[1], **Hikaru Shindo**[1], **Patrick Schramowski**[1,2,8,9],
**Kristian Kersting**[1,2,3,8]
[1]TU Darmstadt  [2]hessian.AI  [3]Lab1141  [4]Aleph Alpha Research  [5]MPI-Inf, SIC
[6]Meta FAIR  [7]Adobe Applied Research  [8]DFKI  [9]CERTAIN, Germany

## ABSTRACT

Large language models (LLMs) excel at generating fluent text, but their internal reasoning remains opaque and difficult to control. Sparse autoencoders (SAEs) make hidden activations more interpretable by exposing latent features that often align with human concepts. Yet, these features are fragile and passive, offering no mechanism for systematic reasoning or model control. To address this, we introduce **ACTIVATIONREASONING** (**AR**), a framework that embeds explicit logical reasoning into the latent space of LLMs. It proceeds in three stages: (1) *Finding latent representations*, first, latent concept representations are identified (*e.g.* via SAEs) and organized into a dictionary; (2) *Activating propositions*, at inference time, AR detects activating concepts and maps them to logical propositions; and (3) *Logical reasoning*, applying logical rules over these propositions to infer higher-order structures, compose new concepts, and steer model behavior. We evaluate AR on multi-hop reasoning (PrOntoQA), abstraction and robustness to indirect concept cues (Rail2Country), reasoning over natural and diverse language (ProverQA), and context-sensitive safety (BeaverTails). Across all tasks, AR scales robustly with reasoning complexity, generalizes to abstract and context-sensitive tasks, and transfers across model backbones. These results demonstrate that grounding logical structure in latent activations not only improves transparency but also enables structured reasoning, reliable control, and alignment with desired behaviors, providing a path toward more reliable and auditable AI. Code and Dataset available at
https://github.com/ml-research/ActivationReasoning

## 1 INTRODUCTION

Large language models (LLMs) demonstrate remarkable abilities in semantic disambiguation, knowledge retrieval, and generative tasks (Brown et al., 2020; OpenAI, 2023). Yet, their internal activations are distributed and entangled, with multiple unrelated features encoded in overlapping representational dimensions. This phenomenon, known as superposition, obscures conceptual representations and limits both interpretability and intervention Marcus (2020); Bender et al. (2021); Elhage et al. (2022). More fundamentally, autoregressive transformers lack explicit propositional structure, making systematic reasoning, compositional generalization, and rule enforcement difficult (Elhage et al., 2022; Bommasani et al., 2021; Woydt et al., 2025; Delfosse et al., 2025). These challenges extend beyond transparency to robustness, controllability, and safe deployment in real world (Gabriel, 2020; Wei et al., 2022). Mechanistic interpretability, and particularly sparse autoencoders (SAEs), promises remedy by surfacing latent features that often align with human-recognizable, monosemantic concepts (Bricken et al., 2023; Templeton et al., 2024). However, SAEs alone remain limited; features can still be polysemous, contextually unstable, or overly low-level (Leask et al., 2025; Härle et al., 2025; Peng et al., 2025). Critically, SAEs lack mechanisms for composition and higher-order reasoning.

Formal logic offers explicit compositionality, well-defined inference rules, and transparency in deriving conclusions (Lloyd, 1984; Russell & Norvig, 2009; Shapiro & Kouri Kissel, 2024). Yet,

---

[*]Equal contribution. Correspondence: {helff, ruben.haerle}@cs.tu-darmstadt.de
[†]Work conducted while at TU Darmstadt/hessian.AI/Lab1141.

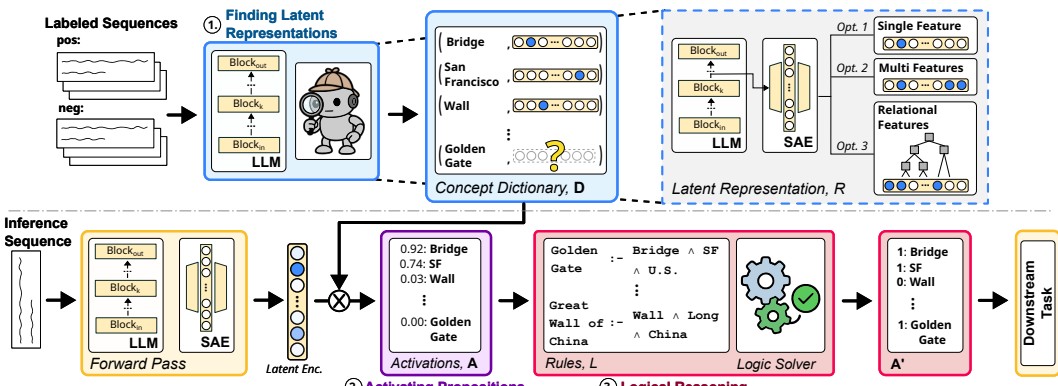

Figure 1: Overview of ACTIVATIONREASONING. AR performs logical reasoning over LLM activations in three stages: (1) Finding latent representations, where concepts are identified in the SAE latent space and stored in a concept dictionary using single, multi, or relational feature representations; (2) Activating propositions, where token-level activations are detected during inference to form an activation matrix $A$; and (3) Logical reasoning, where pre-defined rules are applied over $A$ to infer new higher-order structures, compose new propositions, yielding an enriched matrix $A'$. The structured activations can then be used for downstream transparency and control.

logic-based reasoning presupposes discrete propositional units, whereas LLMs rely on continuous, superposed representations that obscure such structure (Marcus, 2020; Bender et al., 2021; Stammer et al., 2024). While this mismatch has long hindered direct integration, recent progress with SAEs provides a concrete foothold by inducing monosemantic features that approximate discrete propositional units required for symbolic reasoning. To this end, we propose ACTIVATIONREASONING (AR), a framework that embeds explicit logical reasoning into the latent space of LLMs. AR operates in three stages (see Fig. 1): (1) Finding latent representations, where atomic concepts are grounded in SAE-derived features; (2) Activating propositions, which monitors their activations during inference; and (3) Logical reasoning, which composes these propositions into higher-order concepts through user-defined rules. This pipeline extends SAEs from feature extraction to structured reasoning and control. As illustrated in Fig. 2, AR can recover composite concepts absent from the SAE space (*e.g.*, *Golden Gate Bridge* from *Bridge*, *San Francisco*, and *US*); disambiguate polysemous features such as different uses of *red*; enforce safety constraints directly in latent space; and perform multi-step reasoning. On the analysis side, this enables fine-grained inspection of model behavior, such as tracing failure cases to specific activations, probing compositional generalization, or evaluating adherence to safety rules. On the control side, AR supports direct interventions during inference by amplifying or suppressing activations to steer the model toward desired behaviors.

We evaluate AR across four complementary settings. On PrOntoQA (Saparov & He, 2023; Saparov et al., 2023), AR successfully solves deductive multi-hop reasoning tasks without degrading with task complexity. On our novel Rail2Country dataset, it generalizes from explicit lexical concepts to meta-level descriptions expressed through similes and world knowledge. On ProverQA (Qi et al., 2025), which introduces naturalized logical reasoning tasks, AR demonstrates its ability to handle complex reasoning scenarios that reflect real-world linguistic variability. On BeaverTails (Ji et al., 2023), it captures abstract and context-sensitive safety concepts in real-world text. In all tasks, AR substantially outperforms the baselines, improving model reliability and downstream reasoning.

Our contributions are threefold: (i) we introduce ACTIVATIONREASONING (AR), a framework that embeds logical reasoning into the latent space of LLMs by treating SAE features as propositions and composing them through explicit rules; (ii) a new benchmark for reasoning over meta-level concept descriptions, called Rail2Country; and (iii) we demonstrate through experiments on reasoning, meta-concept generalization, and safety that latent activations serve as a viable substrate for structured logical reasoning, enhancing transparency, robustness, and control in LLMs.

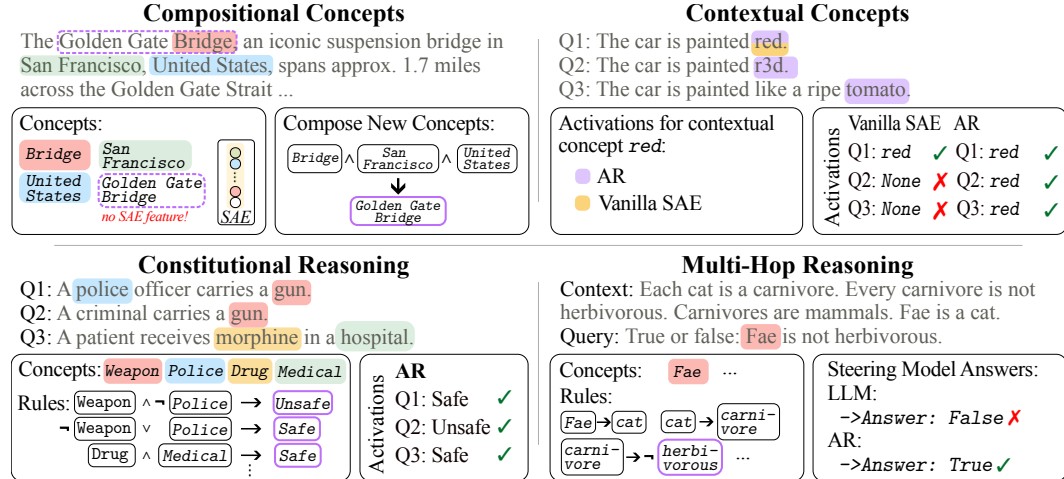

Figure 2: AR applies logical rules on latent activations to enable downstream reasoning, abstraction, and control. AR composes missing concepts from existing SAE features (*top left*), aids robustness to spelling errors or similes (*top right*), implements rules for safeguarding at the neural level (*bottom left*), and solves multi-hop reasoning tasks via logical inference in the latent space (*bottom right*).

## 2 RELATED WORK

Our work touches upon three major research areas: (i) concept-level interpretability of neural representations, (ii) compositional reasoning, and (iii) applications in safety, alignment, and controllability.

**Neural Concept-level Interpretability.** A large body of work has sought to make neural activations more interpretable by grounding them in human-understandable concepts. SAEs (Bricken et al., 2023; Templeton et al., 2024) have recently emerged as a scalable approach, particularly for large models, producing sparse latent features that frequently correspond to semantically meaningful concepts. Follow-up work has revealed both their promise and limitations, highlighting polysemy, context dependence, and difficulties in capturing higher-level abstractions (Leask et al., 2025; Härle et al., 2025; Peng et al., 2025). Hierarchical extensions such as Matryoshka SAEs (Bussmann et al., 2025) attempt to capture layered or compositional features but lack principled control over abstraction levels. Earlier concept-based approaches include supervised methods such as Concept Activation Vectors (CAV) (Kim et al., 2018), as well as Concept Bottleneck Models (CBM) (Koh et al., 2020; Delfosse et al., 2024; Steinmann et al., 2024) and their extensions (Havasi et al., 2022; Shang et al., 2024), which separate prediction into concept detection and downstream classification. More recent work reduces supervision, using pre-trained vision language models such as CLIP (Bhalla et al., 2024; Yang et al., 2023; Oikarinen et al., 2023) or even unsupervised discovery methods (Ghorbani et al., 2019; Stammer et al., 2024; Schut et al., 2025). The majority of these approaches, from interpretable-by-design to mechanistic interpretability, typically treat concepts in isolation. Only a few provide a formal mechanism to *compose* concepts into higher-level abstractions or to disambiguate polysemous features. AR builds on this line of work by grounding logical propositions in latent SAE features and introducing a framework for the logical composition and reasoning over these propositions.

**Compositionality and Reasoning.** Neuro-symbolic AI (Garcez et al., 2015; Sarker et al., 2021; Kautz, 2022; Marra et al., 2024) has long sought to integrate the strengths of neural representations with the structure of symbolic reasoning. Approaches such as DeepProbLog (Manhaeve et al., 2018), SLASH (Skryagin et al., 2021; 2023), and differentiable inductive logic programming frameworks such as $\alpha$ILP (Shindo et al., 2023) and NEUMANN (Shindo et al., 2024) embed logical constraints into differentiable models, enabling joint learning of concepts and rules. Other systems, such as neuro-symbolic concept learners (Stammer et al., 2021; Barbiero et al., 2023; 2024; Wüst et al., 2024), propose integration of concept-level representations with reasoning modules for improved human-AI interactions and explainability. In parallel, work on reasoning LLMs (OpenAI, 2023; Yang et al., 2025; DeepSeek-AI, 2025) has demonstrated remarkable gains by scaling test-time compute. However, their reasoning remains unreliable and difficult to audit (Shojaee et al., 2025; Xie et al., 2024). Our approach differs by explicitly imposing a logical layer on top of latent activations, rather

than requiring differentiable end-to-end training. By defining rules over SAE-based features, AR enables interpretable and auditable reasoning within the latent space of large AI models, bridging the gap between latent activations and symbolic inference.

**Applications in Alignment, Safety, and Knowledge Integration.** Concept-level interpretability is particularly relevant for safe and controllable AI (Kambhampati et al., 2022; Stammer, 2025). Previous work has used interpretable latent features for controllable generation and steering (Turner et al., 2023; Liu et al., 2024), as well as for safety monitoring through moderation systems (Markov et al., 2023; Friedrich et al., 2023; Brack et al., 2023) or direct interventions on safety-relevant features (Härle et al., 2025). More broadly, structured knowledge integration has been shown to improve reliability in domains such as factual grounding and rule-based agents (Creswell et al., 2022; Yu et al., 2023). AR extends these directions by enabling explicit logical constraints and abstractions to be applied directly within the latent space, providing both transparency and controllability for downstream tasks such as safe model steering, interpretable classification, and multi-hop reasoning.

## 3 ACTIVATIONREASONING

ACTIVATIONREASONING (AR) enables logical reasoning directly within the latent activations of LLMs (Fig. 1), yielding interpretable, controllable, and robust behavior in downstream tasks. The process has three stages: (i) **Finding latent representations**, where concept representations are identified in the LLM's latent space and collected in a dictionary; (ii) **Activating propositions**, where token-wise activations are detected at inference time and treated as propositional units; and (iii) **Logical reasoning**, where rules are applied over the activated propositions to infer higher-order structures, compose new concepts, and steer model behavior. As a running example, we infer the concept *Golden Gate Bridge*, which is not initially present in the dictionary but can be derived from the joint activation of three base concepts: *Bridge*, *San Francisco*, and *USA*.

### 3.1 FINDING LATENT REPRESENTATIONS

The first stage of AR (*cf.* Fig. 1, top half) constructs a set of interpretable concepts grounded in the latent space of the LLM. We first introduce a formal definition of a concept and then explain how the associated representations are constructed.

**Formalizing Concepts.** We formally define a concept $c$ as a tuple $(n_c, r_c, \tau_c)$. Let $\mathcal{L} \subseteq \mathbb{R}^d$ denote the latent space of concept codes produced by a fixed encoder $E$ applied to the model's hidden states. For each token $t$ with hidden state $h_t$, we obtain a latent code via the encoder $\ell_t = E(h_t) \in \mathcal{L}$.

Concretely: (i) $n_c$ is a semantic identifier for the concept (e.g., "Bridge"). (ii) $\tau_c \in \mathbb{R}_{\geq 0}$ is a soft activation threshold that determines when the concept is considered active. (iii) $r_c : \mathcal{L} \to \mathbb{R}_{\geq 0}$ is the concept's latent representation in this space. Implemented as a function, it maps latent codes $\ell_t \in \mathcal{L}$ to a activation score $a(c, t) := r_c(\ell_t) \in \mathbb{R}_{\geq 0}$. The collection of concepts used by AR forms a concept dictionary $\mathcal{D} = \{c_i\}_{i=1}^{N}$ containing the building blocks for subsequent logical reasoning.

**Toward interpretable activations.** Raw LLM activations are highly entangled and subject to superposition, encoding multiple features in overlapping dimensions (Elhage et al., 2022), which obstructs recovery of monosemantic representations (Park et al., 2023). Logical reasoning, however, requires such representations as propositional building blocks. Thus, we project activations into a sparse and more interpretable feature space using SAEs (Bricken et al., 2023), which promote monosemanticity by aligning individual dimensions with distinct, human-interpretable concepts. The framework remains method-agnostic and can directly integrate future advances in interpretability.

**Types of latent representations.** Building on SAE features, we propose three forms of latent representation $r \in \{\mathcal{R}_{\text{single}}, \mathcal{R}_{\text{multi}}, \mathcal{R}_{\text{relation}}\}$ (*cf.* Fig. 1, top right). (1) A *Single-feature representation* ($\mathcal{R}_{\text{single}}$) ties a concept to one SAE feature, reflecting the common assumption that some features can align closely with human-interpretable notions (Bricken et al., 2023). (2) A *Multi-feature representation* ($\mathcal{R}_{\text{multi}}$) instead associates a concept with a set of $k$ SAE features. This relaxes the strict monosemantic assumption, acknowledging that features often capture only fragments of a concept or exhibit polysemy (Leask et al., 2025; Härle et al., 2025). Here, features contribute independently through weighted aggregation; no interactions between them are modeled. (3) A *Relational-feature representation* ($\mathcal{R}_{\text{relation}}$) defines a concept by a set of relations among SAE

features. In practice, we instantiate this via a shallow decision tree that applies different thresholds and conditions along different branches, balancing expressiveness with interpretability. Unlike $\mathcal{R}_{\text{multi}}$, this can express structured interactions such as conjunctions (requiring *both* slurs *and* stereotypes) or context-dependent exclusions (filtering educational mentions). Simple concepts like *Bridge* or *USA* can be captured by $\mathcal{R}_{\text{single}}$ or $\mathcal{R}_{\text{multi}}$, while abstract or contextual notions benefit from $\mathcal{R}_{\text{relation}}$. *E.g.*, the broader concept *hate* may be recognized if features linked to *slurs* or *demeaning stereotypes* are activated, while benign uses of similar words are filtered out. Such patterns require that multiple cues co-occur while others are excluded (e.g., filtering non-toxic educational usage). These structured dependencies are characteristic of $\mathcal{R}_{\text{relation}}$ and cannot be captured by linear aggregation alone.

**Concept extraction.** Concepts can be obtained either manually or automatically. In the *manual* case, a human expert defines $r_c$ by assigning SAE encoder features $(\text{SAE}_E)$ to concepts. For example, features consistently activating for the notion of a *Bridge* can be directly assigned. This direct assignment is used, however, only when a feature is clearly monosemantic, and its link to a concept is unambiguous; in practice, most concepts are constructed via the automatic procedure described next, which scales naturally and does not require manual inspection.

In the *automatic* setting, representations are induced from data with token-level labels for the concepts. The goal is to find latent representations that act as reliable "signatures" for concepts. For instance, to find the representation for *Bridge*, we identify SAE features that most effectively differentiate tokens labeled as *Bridge* from those that are not. Given the hidden state $h_t$ for a token $t$ and its sparse code $l_t = \text{top-}k(\text{SAE}_E(h_t))$, we derive the representation $r_c$ for a concept $c$ with label $y_{c,t}$ as

$$r_c = \begin{cases} \arg\max \ \left(\mathbb{E}[l_t \mid y_{c,t} = 1] - \mathbb{E}[l_t \mid y_{c,t} = 0]\right) & \text{for} \quad \mathcal{R}_{\text{single}} \\ \text{top-}k \ \left(\mathbb{E}[l_t \mid y_{c,t} = 1] - \mathbb{E}[l_t \mid y_{c,t} = 0]\right) & \text{for} \quad \mathcal{R}_{\text{multi}} \\ \text{decision tree induced from } \left(l_t, y_{c,t}\right) & \text{for} \quad \mathcal{R}_{\text{relation}} \ . \end{cases} \tag{1}$$

Here, $\mathbb{E}[\cdot \mid y_{c,t}]$ denotes the empirical expectation over token-level sparse codes $l_t$, conditioned on whether a token $t$ is labeled as belonging to concept $c$ ($y_{c,t} = 1$) or not ($y_{c,t} = 0$).

**Thresholding activations.** Each representation $r_c$ is paired with a soft threshold $\tau_c$ to decide when the concept is considered active. In the automatic case, $\tau_c$ is chosen to maximize balanced accuracy for the provided labels $y_{c,t} \in \{0, 1\}$ and scores $a_{c,t}$ as

$$\tau_{c_i} \in \arg\max_{\tau \geq 0} \ \tfrac{1}{2}\left(\text{TPR}_c(\tau) + \text{TNR}_c(\tau)\right) , \tag{2}$$

where $\text{TPR}_c(\tau) = \Pr(a_{c,t} \geq \tau \mid y_{c,t} = 1)$ and $\text{TNR}_c(\tau) = \Pr(a_{c,t} < \tau \mid y_{c,t} = 0)$. Here $a_{c,t}$ denotes the activation score for concept $c$ at token $t$, as formally defined in Sec. 3.2. In the manual setting, where labels are unavailable, the threshold $\tau_{c_i}$ defaults to 0 but can be overridden by a user-specified value.

## 3.2 ACTIVATING PROPOSITIONS

With the concept dictionary $D$ established, this stage tracks activations in the latent space and maps them to atomic propositions for logical reasoning. This is a two-step process. First, we compute raw, token-level activation scores to detect local evidence for each concept. Second, we formalize these activations into propositions by applying thresholds and, where necessary, aggregating them.

**Activation.** At inference, we compute an activation score $a(c, t)$ for each concept $c$ at each token $t$. Given the hidden state $h_t$, we obtain its sparse code $l_t = \text{top-}k_{\text{in}}(\text{SAE}_E(h_t))$. The score $a(c, t)$ is then derived from the latent representation $r_c$ as

$$a(c, t) = \begin{cases} l_t[r_c] & \text{if } r_c \in \mathcal{R}_{\text{single}} \\ \sum w \, l_t[r_c] & \text{if } r_c \in \mathcal{R}_{\text{multi}} \\ r_c(l_t) & \text{if } r_c \in \mathcal{R}_{\text{relation}} \ . \end{cases} \tag{3}$$

For $\mathcal{R}_{\text{single}}$, the score corresponds to the value of a single SAE feature (*cf.* Fig. 1, purple box). For $\mathcal{R}_{\text{multi}}$, evidence is aggregated from multiple indices using a weighting vector $w$ (*e.g.*, log decay), normalized to sum to one, *i.e.*, forming a convex combination. For $\mathcal{R}_{\text{relation}}$, the score is obtained by traversing the decision tree $r_c$ on $l_t$, producing a probability of concept presence.

**From Activations to Propositions.** To make activations accessible for reasoning, we collect them into the *activation matrix* $A$ that serves as weighted evidence for concept activity. We provide (i) token-level activations $A_{\text{local}} \in \mathbb{R}^{|D| \times |S|}$, and (ii) sequence-level activations $A_{\text{global}} \in \mathbb{R}^{|D|}$ for each concept $c \in D$ and token $t \in S$:

$$A_{local}[c,t] \ = \ \max(a_{c,t} - \tau_c, 0), \quad A_{global}[c] \ = \ \max(\text{Agg}_{t \in S}\, a_{c,t} - \tau_c, 0) \ . \qquad (4)$$

Here, $\text{Agg}$ denotes an aggregation operator (mean by default), and $\tau_c$ the concept-specific soft threshold. Intuitively, $A_{local}$ provides fine-grained token-level evidence of concept activity, whereas $A_{global}$ reflects higher-level semantic meaning throughout the sequence. Individual words may only weakly signal a concept, while aggregated evidence over a span reveals a more robust semantic direction. For the subsequent reasoning stage, we can now view each concept as a logical proposition with its weighted evidence score encoded into the activation matrix, either locally or globally.

Consider the following input sequence: *"After the fourth failed selfie on the Golden Gate Bridge in SF, USA, she said, 'Well, that went well.'"* At the token level, AR assigns high activations to the base concepts at their respective positions, *e.g.*, $A_{local}[Bridge, t]$, $A_{local}[San\ Francisco, t]$, and $A_{local}[USA, t]$. These activations constitute *local evidence*, directly grounded in individual tokens. By contrast, *global evidence* captures abstract semantics that emerge only at the sequence level: here, the ironic clause "Well, that went well." activates the higher-order concept *sarcasm* with $A_{global}[sarcasm]$. This illustrates how local activations provide explicit, token-level signals, while sequence-level aggregation enables the detection of more abstract semantic notions that are not necessarily attributable to any single token.

### 3.3 Logical Reasoning

Given the activation matrix $A$, we apply logical rules $L$ to infer new propositions (*cf.* Fig. 1, red box). Rules are user-defined (*cf.* Fig. 2) and use standard propositional operators (implication, conjunction, disjunction, negation). For example, a rule may encode semantic and compositional relations, *e.g.*, $Bridge \wedge San\ Francisco \wedge USA \rightarrow Golden\ Gate\ Bridge$.

**Reasoning.** Having established a set of user-defined rules, reasoning proceeds via *forward chaining*. Starting from the initial activation matrix $A$ of logical propositions, inference rules are applied iteratively to derive new conclusions, until a fixed point is reached where no further inferences can be drawn (Russell & Norvig, 2009). We formalize the explicit semantics in App. D. In practice, rules may operate over $A_{\text{local}}$ and $A_{\text{global}}$. The former provides evidence for explicit concepts such as *Bridge*, *San Francisco*, and *USA* at specific tokens, while the latter captures more abstract notions in the sequence, such as *sarcasm*. For reasoning, we feed discretized activations $A$ into the rule engine (*e.g.*, $A_{\text{local}}[Bridge, 15] = 0.9$ is treated as the Boolean proposition *Bridge*). Applying the compositional rule above to the evidence allows us to infer the higher-level proposition *Golden Gate Bridge*, even if no SAE feature directly encodes this concept.

**Enriched matrix.** We denote the resulting enriched matrix of logical propositions as $A'$ (*cf.* Fig. 1, bottom right). In contrast to the raw activations $A$, which contain only directly activated concepts, $A'$ integrates both detected and inferred propositions, providing a structured and interpretable representation that can be leveraged for downstream analysis and control.

### 3.4 Integration into Downstream Tasks

For *analysis*, $A'$ allows fine-grained inspection of model behavior, such as tracing failure cases to specific concept activations (*cf.* Sec. 4, Rail2Country), probing compositional generalization (*cf.* Fig. 2) or evaluating adherence to safety constraints (*cf.* Sec. 4, safety experiments).

On the other hand, in terms of *control*, $A'$ supports interventions during inference by activating or suppressing model activations using the concept representations to steer the model toward desired behaviors (*cf.* Sec. 4, PrOntoQA). For steering, we build upon decoder-based model steering (Härle et al., 2025) to bias the model during generation towards activated concepts from the final $A'$. For example, if we want to steer towards the concept *red*, the associated latent representation $r_{(\text{red})} \in \mathcal{R}_{\text{multi}}$ is retrieved from our concept corpus $C$ and used to extract the corresponding SAE decoder weights ($SAE_D$) for *red*. Subsequently, we apply weighting vector $w$ (*e.g.*, log decay) and normalize with respect to the latent activations of the model. The steering factor $\alpha \in \mathbb{R}$ indicates the amount of

promotion or suppression of the concept $c$. Finally, we update the model's latent activations $h$ as

$$h' = h + \alpha \cdot \frac{(SAE_D[r_c] \times w) \times \|h\|_2}{\|SAE_D[r_c]\|_2} \ . \tag{5}$$

Overall, structuring AR into activations, logical reasoning, and downstream integration yields a modular pipeline: concept representations $r_c$ provide interpretability, logical rules $L$ enable compositional reasoning, and the enriched concept matrix $A'$ supplies actionable signals for alignment and control.

## 4 EXPERIMENTAL EVALUATIONS

In our evaluations, we first investigate whether AR can perform reasoning directly on latent activations and whether it does so robustly as task complexity scales (PrOntoQA). Next, we investigate whether AR generalizes beyond explicit lexical cues to meta-level concept descriptions expressed through similes and compositional object semantics (Rail2Country). Then, we move to reasoning tasks with natural and diverse language that combine real-world linguistic variability with complex logical reasoning (ProverQA). Finally, we examine its ability to reason over abstract safety concepts in real-world text, where categories are vague and context-dependent (BeaverTails).

**Experimental Setup.** We evaluate AR on two backbone models, Llama-3.1-8B (AI@Meta, 2024) with EleutherAI's SAE (EleutherAI, 2024) attached after layer 23, and Gemma-2-9B (Team, 2024) with a Gemma-Scope SAE (Lieberum et al., 2024) attached after layer 20. All experiments use greedy decoding. For evaluation, we apply $\mathcal{R}_{\text{multi}}$ in the first three benchmarks and compare $\mathcal{R}_{\text{single}}$, $\mathcal{R}_{\text{multi}}$, and $\mathcal{R}_{\text{relation}}$ in the fourth. We report results against several baselines: the same backbones without AR, larger instruction-tuned models (Llama-3.1-70B and Gemma-2-27B), commercial models (GPT-4o (OpenAI, 2023)), and the reasoning model DeepSeek-R1-Distill-Llama-8B (DeepSeek-AI, 2025). Additionally, we provide comparison to further instruction-tuned and chain-of-thought baselines in App. E and analysis of runtime efficiency in App. B. Baseline LLMs generate answers directly, whereas AR performs reasoning in the activation space and steers the downstream generation. We provide detailed descriptions of the datasets in App. A and the hyperparameters in App. H. We also include an ablation study on the choice of SAE placement, demonstrating that AR remains invariant to this factor (*cf.* App. G).

**Multi-Hop Reasoning on Latent Activations.** We begin by investigating to what extent AR can enhance reasoning. To do so, we evaluate on PrOntoQA (Saparov & He, 2023; Saparov et al., 2023), a popular question-answering dataset that tests deductive reasoning and scales complexity by increasing the number of reasoning hops (1, 3, or 5). In PrOntoQA, models are provided with a *context* containing a set of rules and a *query* that has to be proven or disproven. We initialize AR with $\mathcal{R}_{\text{multi}}$ using the PrOntoQa train set and encode the context into the rules $L$.

Results on PrOntoQA testset are summarized in Tab. 1 (left). As expected, the 8B and 9B base models perform poorly and remain close to chance, often failing even the simplest 1-hop questions. Larger instruction-tuned models (Llama-3.1-70B, Gemma-2-27B, and GPT-4o) achieve strong performance on the 1-hop setting, but their accuracy drops sharply as the number of reasoning hops increases, revealing limitations to generalize to longer and more complex reasoning scenarios. The reasoning model, DeepSeek-R1-Distill-8B, is also subject to degradation as task complexity scales. In contrast, AR consistently solves most problems across hop lengths, explicitly inferring logical conclusions within activation space and steering the model to correct answers. This yields accuracies above 93% at 1, 3, and 5 hops. Importantly, unlike all baselines, AR maintains its performance as task complexity scales, showing that the compositional rules it encodes generalize robustly to longer and more complex reasoning chains. Furthermore, gains transfer across different backbone families.

**Reasoning over Meta-level Concept Descriptions.** We next examine whether AR can generalize beyond explicit lexical mentions to abstract, meta-level descriptions. Prior work shows that SAEs reliably recover consistent features when concepts are explicitly referenced (Bricken et al., 2023; Templeton et al., 2024); for instance, the token *red* tends to activate a stable subset of SAE features across different contexts. However, when concepts are expressed indirectly through similes or descriptive proxies (*e.g.*, *colored like a tomato* as a substitute for *red*), the resulting activations become weaker and distributed, introducing noise into the latent representation (Leask et al., 2025).

To study this setting, we introduce **Rail2Country (R2C)**, a newly introduced benchmark for reasoning over meta-level concept descriptions, inspired by earlier train-based reasoning tasks (Michalski, 1980;

Table 1: **Reasoning on latent activations.** Exact-match accuracy on PrOntoQA (1–5 hop reasoning), Rail2Country (Mono with explicit concepts; Meta with similes, *e.g.*, 'red' → 'like a tomato'), and ProverQA (linguistically diverse reasoning tasks across difficulty levels). AR consistently boosts multi-hop reasoning, remains robust as task complexity scales, and generalizes to natural and diverse language–outperforming its baselines, and even other reasoning/larger instruction-tuned (it) LLMs.

| Model | PrOntoQA (↑%) | | | Rail2Country (↑%) | | ProverQA (↑%) | | |
|---|---|---|---|---|---|---|---|---|
| | 1 Hop | 3 Hops | 5 Hops | R2C-Mono | R2C-Meta | Easy | Medium | Hard |
| Llama3.1 8B | 51.0 | 50.8 | 50.3 | 41.0 | 29.7 | 43.6 | 33.6 | 36.8 |
| w/ AR (our) | **95.0** (+44.0) | **95.6** (+44.8) | **95.3** (+45.0) | **74.7** (+33.7) | **62.7** (+33.0) | **92.8** (+49.2) | **91.0** (+57.4) | **70.8** (+34.0) |
| Gemma2 9B | 48.5 | 47.5 | 47.9 | 35.3 | 25.7 | 39.4 | 29.8 | 25.8 |
| w/ AR (our) | **93.5** (+45.0) | **93.5** (+46.0) | **93.5** (+45.6) | **93.7** (+58.4) | **86.0** (+60.3) | **94.0** (+54.6) | **91.4** (+61.6) | **69.6** (+43.8) |
| **Other Baselines** | | | | | | | | |
| Llama3.1 70B it | 96.2 | 66.7 | 62.1 | 68.3 | 33.3 | 74.8 | 58.8 | 41.0 |
| Gemma2 27B it | 91.3 | 77.6 | 73.9 | 61.0 | 45.0 | 74.8 | 69.0 | 46.8 |
| GPT-4o | 97.3 | 74.4 | 66.4 | 82.7 | 69.0 | 81.0 | 65.4 | 46.4 |
| DeepSeek-R1-8B | 80.0 | 76.0 | 64.4 | 83.7 | 60.0 | 65.6 | 58.6 | 44.2 |

Mitchell, 1997; Helff et al., 2024; 2025). Each instance consists of a *context* that lists countries with their flag color sequences and a *query*, a short train description whose sequence of car colors must be mapped to the corresponding country (*e.g.*, red–white–blue → *France*). R2C comes with two versions. In **R2C-Mono**, colors are explicitly stated in the trains' descriptions, producing clear SAE activations (Bricken et al., 2023; Templeton et al., 2024). In **R2C-Meta**, explicit color mentions are replaced with similes (*e.g.*, 'colored like a tomato' for red). In this way, a color cue is entangled with object semantics ('tomato'), contextual phrasing ('like'), and world knowledge (tomatoes are red). Such descriptions are expected to distribute activation across multiple SAE features (Elhage et al., 2022). Further details and examples are provided in App. A.2.1. For evaluation, we initialize AR with $\mathcal{R}_{\text{multi}}$ using the R2C-Mono train set and encode the context rules into $L$.

Results are summarized in Tab. 1 (middle). In R2C-Mono, baseline LLMs achieve only 41%/35% (Llama/Gemma), failing even when concepts are explicitly stated. In contrast, AR substantially improves reasoning, reaching 75%/94% (Llama/Gemma), with AR (Gemma) outperforming not only its larger counterpart (Gemma-27B) but also surpassing GPT-4o and the reasoning model DeepSeek-R1-8B. In R2C-Meta, the task becomes more challenging as models must resolve implicit cues by integrating contextual phrasing and world knowledge. Performance of the small baselines deteriorates further to 30% and 26%, and even larger instruction-tuned models show sharp drops. GPT-4o and DeepSeek-R1-8B remain comparatively stronger, but still degrade and do not exceed 70%, despite using far more parameters and test time tokens. Even though implicit descriptions scatter activations across the SAE feature space, AR remains relatively robust, with AR (Gemma) achieving the best overall performance (86%). To understand this behavior, we separately evaluate concept detection in the latent space (see App. F). While vanilla SAEs collapse completely (0% accuracy), AR reliably recovers most implicit concepts, explaining its strong performance.

**Towards Real-World Reasoning Scenarios.** Next, we extend our evaluation to ProverQA (Qi et al., 2025), a benchmark of LLM-generated problems that present naturalized reasoning tasks which reflect real-world linguistic diversity. It has three complexity tiers, namely, easy, medium, and hard. Each task provides a *context* of facts and rules as well as a *query* that is to be proven true, false, or uncertain. For evaluation, we initialize AR with $\mathcal{R}_{\text{multi}}$ using the set of available propositions, and encode the context rules into $L$. Results on ProverQA are summarized in Tab. 1 (right). We observe that baseline LLMs perform poorly. Larger instruction-tuned models and GPT-4o achieve stronger results on the easy tier but deteriorate on medium and hard problems, and the reasoning model DeepSeek-R1-8B follows a similar trend. In contrast, AR yields substantial gains across all tiers, AR (Llama) reaches 93%/91%/71% (easy/medium/hard) and AR (Gemma) 94%/91%/70%, consistently outperforming both the much larger LLMs and reasoning models. On the hard tier, only AR maintains solid performance near 70%, while all other models fall below 50%. This highlights its robustness to linguistically diverse and complex reasoning tasks and demonstrates that structured reasoning in latent space generalizes effectively to natural, increasingly difficult scenarios.

**Reasoning over Abstract Safety Concepts.** To test whether AR supports controllability in real-world use cases, we evaluate on the BeaverTails dataset (Ji et al., 2023). This dataset reflects the diversity

Table 2: **Safety Evaluation.** Balanced accuracy (%) across 14 BeaverTails safety dimensions, with deltas indicating improvements over Base SAE. Relational AR (Llama3.1 8B) achieves the best overall performance, while multi-feature (5) AR also provides strong gains over flat SAE features.

| Method | Animal | Child | Politics | Discrim. | Drugs | Finance | Hate | |
|---|---|---|---|---|---|---|---|---|
| Base SAE | 64.3 | 57.4 | 53.6 | 52.2 | 61.1 | 63.0 | 52.9 | |
| **AR Single** | 85.8 (+21.5) | 91.3 (+33.9) | 66.1 (+12.5) | 76.2 (+24.0) | 78.9 (+17.8) | 80.6 (+17.6) | 74.8 (+21.9) | |
| **AR Multi** | 92.7 (+28.4) | **95.3** (+37.9) | 76.5 (+22.9) | 79.8 (+27.6) | 81.7 (+20.6) | 81.7 (+18.7) | 79.7 (+26.8) | |
| **AR Relation** | **94.7** (+28.7) | 94.4 (+35.0) | **81.2** (+24.7) | **84.9** (+28.2) | **91.8** (+31.2) | **85.6** (+23.0) | 70.7 (+23.7) | |
| | Misinfo. | Unethical | Privacy | Self-harm | Sexual | Terrorism | Violence | **Overall** |
| Base SAE | 50.7 | 51.2 | 58.5 | 73.6 | 68.2 | 52.2 | 51.5 | 57.9 |
| **AR Single** | 51.2 (+0.5) | 59.3 (+8.1) | 85.1 (+26.6) | 79.5 (+5.9) | 91.1 (+22.9) | 52.2 (0.0) | 55.6 (+4.1) | 73.4 (+15.5) |
| **AR Multi** | 51.1 (+0.4) | 66.3 (+15.1) | 89.3 (+30.8) | 78.9 (+5.3) | 90.8 (+22.6) | 72.4 (+20.2) | 72.5 (+21.0) | 79.2 (+21.3) |
| **AR Relation** | **62.0** (+9.2) | 64.8 (+17.2) | 86.8 (+28.7) | **89.5** (+16.2) | **92.0** (+24.5) | **86.0** (+34.9) | **78.0** (+27.7) | **83.0** (+25.2) |

and nuance of real-world safety classification, spanning 14 safety-relevant dimensions, including *Animal Abuse*, *Discrimination*, *Hate Speech*, and *Self-Harm* (for further details, see App. A.4.1). We use a subset of 3.7k training examples to select the highest activating feature for the base SAE, initialize AR's concept representations, and encode logic rules into $L$ that map BeaverTails' safety categories to *Unsafe* (*e.g.*, *Terrorism* → *Unsafe*).

Results on the 3k test set, summarized in Tab. 2, measure whether the unsafe concept correctly activates in $A'$. The base SAEs struggle with context-sensitive and abstract safety categories, while AR produces consistent improvements, with hierarchical concepts achieving the strongest overall performance. However, some categories, such as *Misinformation Regarding Ethics, Laws, and Safety* and *Non-Violent Unethical Behavior*, remain especially challenging. Here, the base SAE performs close to random guessing, indicating the absence of strongly aligned latent features. Since AR is based on SAE features, its performance is constrained by their representational quality. Overall, however, the results show that AR produces more robust concept representations and substantially outperforms base SAEs, particularly on abstract and context-sensitive safety concepts.

## 5  DISCUSSION

Our results show that by grounding propositions in SAE-derived features, AR turns latent activations into a substrate for logical reasoning, enhancing LLM abilities across several dimensions. Across all four experiments, AR not only improves over its 8B/9B backbones but also surpasses much larger instruction-tuned models, proprietary undisclosed systems like GPT-4o, and even reasoning models like DeepSeek-R1-Distill-Llama-8B. First, in compositional reasoning (*cf.* PrOntoQA), AR enriches concept representations by composing new high-level concepts and applying deductive inference, maintaining accuracies above 93% across 1–5 hops, while all baselines degrade with complexity. Second, for reasoning over implicit or noisy cues (*cf.* R2C), AR remains robust when concepts are expressed indirectly through similes, where both small and large LLMs deteriorate, and vanilla SAEs collapse entirely. Third, on more realistic reasoning scenarios (*cf.* ProverQA), AR generalizes effectively to linguistically diverse problems, retaining strong performance even at the hard tier where all other models fall below 50%. Fourth, in abstract and context-sensitive safety tasks (*cf.* BeaverTails), AR recovers abstract notions and encodes constraints directly into the LLM's latent space. Taken together, these findings demonstrate that structured reasoning in activation space provides a model-agnostic mechanism that not only improves reasoning performance but also increases transparency and control across synthetic, semi-natural, and real-world settings. Notably, AR achieves these improvements without significant overhead, while reasoning and CoT variants are both slower and less accurate (App. B). AR also scales favorably to long contexts (*cf.* Tab. 1). At inference, AR only tracks concept activations for each token, so memory and compute grow linearly with the sequence length and remain independent of the SAE dimensionality. Thus, once concepts are extracted, rule application is purely symbolic and does not degrade with increasing token distance.

**Limitations.** Although our implementation of AR relies on SAEs as a substrate to define logical propositions, the framework itself is not bound to them. Current SAEs offer a practical way to surface

sparse, often interpretable features, but they are not always perfect. Features may be polysemous, overly context-dependent, or fail to capture abstract notions, and AR already mitigates several of these effects through thresholding and multi-feature or relational representations. Pretrained SAEs also allow only limited control over which concepts are discovered and how they are represented. Importantly, AR is not tied to SAEs: any method that extracts sparse or interpretable concept signals from raw activations could be used as a replacement, and future improvements in activation-based concept discovery would transfer directly to our framework. Future work could explore alternative interpretability approaches, such as different autoencoding objectives, clustering methods, or other complementary representation-learning methods, that may yield representations better suited for logical reasoning in certain domains. Second, real-world reasoning challenges can become more diverse than the tasks highlighted in our work. In particular, large-scale datasets involving open-ended reasoning, long-context inference, or knowledge-intensive domains remain unexplored and remain to be evaluated in future work. Third, while the current setup relies on manually or semi-automatically defined rules and concept identification, advances in automated rule induction, probabilistic reasoning, and integration with external knowledge bases can substantially increase the adaptability of our framework. Notably, because AR rule specification is decoupled from model training, rules can be proposed, refined, or revised based on observed data without retraining the LLM or SAE. For instance, LLMs could generate candidate rules directly from natural language descriptions, which could then be validated against evidence and pruned as needed. We also note that the present version of AR is purely deductive; applying forward chaining over binary propositions. Moving beyond purely deductive reasoning, AR's solver-agnostic design also allows integration of probabilistic engines such as SLIPCOVER (Bellodi & Riguzzi, 2013) or ProbFOIL (De Raedt et al., 2015) for handling uncertainty, or inductive/abductive systems (Srinivasan, 2001; Cropper & Morel, 2021) for hypothesis formation and rule discovery from sparse data. Combined with external knowledge bases, these extensions would enable AR to learn domain-specific patterns and generalize beyond manually encoded ontologies, addressing the need for adaptability in open-ended reasoning scenarios.

Finally, AR currently fixes the representation type per concept. Since different concepts benefit from different representation types (single/multi/relational), automatic selection or hybrid combinations represent a natural extension that would improve robustness to polysemantic features and complex feature interactions.

## 6 CONCLUSION

We introduced ACTIVATIONREASONING (AR), a framework that integrates logical reasoning into the latent activation space of LLMs by grounding propositions. This structured bridge between neural activations and symbolic semantics enables explicit composition, disambiguation, and inference over otherwise opaque representations. Beyond interpretability, AR expands the functional scope of LLMs: it turns latent activations into a substrate for structured reasoning, supports direct interventions for control and alignment, and enables auditable mechanisms for safety. In this way, latent spaces become not only more transparent but also more reliable and steerable. We view AR as a step towards unifying the semantic richness of LLMs with the systematic generalization of logic, moving toward models that can reason more robustly and operate in closer alignment with human values.

Future work includes automatic rule discovery, integration with probabilistic or inductive reasoning modules, and automatic selection of concept representation types. Exploring representation-learning approaches beyond pretrained SAEs may yield activation spaces that are better suited for abstract or domain-specific reasoning. While the present work focuses on LLMs, AR is not limited to a specific modality, since it operates fully on concept-level activations rather than a specific architecture or type of activation. Recent progress in multimodal SAEs suggests that similar reasoning could extend to vision or VLMs (Pach et al., 2025; Fel et al., 2025; Cassano et al., 2025), where AR would allow to reason over visual attributes, e.g., enforce safety constraints in image generation, or combine cross-modal cues through rule-based composition. Finally, scaling AR to large-scale open-ended tasks with long contexts and diverse knowledge demands will be key to demonstrating robustness and generality.

## ACKNOWLEDGEMENTS

The authors acknowledge support from the hessian.AI Service Center (funded by the Federal Ministry of Research, Technology and Space, BMFTR, grant no. 16IS22091) and the hessian.AI Innovation Lab (funded by the Hessian Ministry for Digital Strategy and Innovation, grant no. S-DIW04/0013/003), and the Center for European Research in Trusted AI (CERTAIN). This work has also benefited from the BMWE project "EU-SAI: Souveräne KI für Europa," 13IPC040G, and also from early stages of the Cluster of Excellence "Reasonable AI" funded by the German Research Foundation (DFG) under Germany's Excellence Strategy— EXC-3057; funding will begin in 2026. This work was supported by the Priority Program (SPP) 2422 in the subproject "Optimization of active surface design of high-speed progressive tools using machine and deep learning algorithms" funded by the German Research Foundation (DFG). Additional funding was provided by the Aleph Alpha Collaboration Lab 1141. Views and opinions expressed are, however, those of the author(s) only and do not necessarily reflect those of the European Union or the European Health and Digital Executive Agency (HaDEA). Neither the European Union nor the granting authority can be held responsible for them. Further, this work benefited from the ICT-48 Network of AI Research Excellence Center "TAILOR" (EU Horizon 2020, GA No 952215), the Hessian research priority program LOEWE within the project WhiteBox [GA No LOEWE/ 2/13/519/03/06.001(0010)/77], the HMWK cluster projects "Adaptive Mind" and "Third Wave of AI", and from the NHR4CES.

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

# Supplementary Materials

## A  EXPERIMENTAL SETUP

**Models.**  For our experiments, AR is evaluated on two backbone models: Llama-3.1-8B AI@Meta (2024) with the EleutherAI `sae-llama-3.1-8b-64x` EleutherAI (2024) (width 262k) attached at layer 23 [1], and Gemma-2-9B Team (2024) with the `gemma-scope-9b-pt-res-canonical` SAE Lieberum et al. (2024) (width 131) attached at layer 20 [2]. All experiments are conducted with greedy decoding.

### A.1  PRONTOQA EXPERIMENTAL SETUP

For the PrOntoQA experiments, we instantiate AR with multi-feature concept representations $\mathcal{R}_{\text{multi}}$. Unless otherwise noted, activations are aggregated using the mean operator and steering follows the mean-shift update rule.

**Llama.**  For the Llama experiments, concepts were retrieved by searching over word-level units, where all tokens belonging to a candidate word were considered. Building the concept dictionary used the top 10 activating features for representation $r_c$, in the ordering of unique features first. During detection, at most top-$k_{\text{in}} = 2$ features per token and top-$k_{\text{multi}} = 4$ features per concept for $\mathcal{R}_{\text{multi}}$ were retained for computing activations $a_{c,t}$. Steering was applied with factor $\alpha = 0.5$ and restricted to one SAE decoder weight row. *One* steering vector for each *True* and *False* concept were selected manually from the SAE decoder weights.

**Gemma.**  For the Gemma experiments, the same parameters were used to build the concept dictionary (word-level units over all tokens), but with a different ordering strategy: unique-only features were retained. As above, representation search used the top 10 activating features for $r_c$. Detection kept top-$k_{\text{in}} = 7$ features per token and top-$k_{\text{multi}} = 2$ features per concept for $\mathcal{R}_{\text{multi}}$, with an additional threshold $\tau = 14$ imposed on activations before producing the activation matrix $A$. Steering used the same configuration as for Llama, with factor $\alpha = 0.4$, and one SAE decoder weight row. As before, steering vectors for the *True* and *False* concepts were manually selected.

### A.1.1  DATASET DETAILS

We follow the procedure of Hao et al. (2024) and generate the out-of-distribution samples using the publicly available codebase[3]. We divide the dataset as follows: 500 samples are used to extract concepts for detection, while 2000 unseen samples are reserved for evaluation for each subtask (number of hops). The task is binary classification: given a context (ontology description) and a query, the model must decide whether the query is `True` or `False`.

**Context:**  Each zumpus is a wumpus. Lempuses are orange. Zumpuses are brimpuses. Each vumpus is a shumpus. Each wumpus is floral. Dumpuses are lorpuses. Every numpus is fast. Each lempus is a vumpus. Each shumpus is a jompus. Each rompus is muffled. Shumpuses are mean. Each tumpus is shy. Sterpuses are wooden. Each vumpus is snowy. Vumpuses are tumpuses. Each jompus is transparent. Every lempus is a rompus. Each impus is a sterpus. Impuses are not temperate. Dumpuses are not dull. Each zumpus is not fast. Impuses are zumpuses. Shumpuses are impuses. Alex is a dumpus. Alex is a lempus.

**Query:**  Answer True or False: Alex is not fast. Answer:

**Model Inputs:**  The baseline model received the ontology in text form (context) whereas AR the ontology in the ontology in rule form (see below). Both models received the query.

---

**Rules Example**

$$
\begin{aligned}
\text{zumpus} &\rightarrow \text{brimpus}; & \text{zumpus} &\rightarrow \neg\text{fast} \\
\text{zumpus} &\rightarrow \text{wumpus}; & \text{lempus} &\rightarrow \text{orange} \\
\text{lempus} &\rightarrow \text{rompus}; & \text{lempus} &\rightarrow \text{vumpus} \\
\text{vumpus} &\rightarrow \text{shumpus}; & \text{vumpus} &\rightarrow \text{snowy} \\
\text{vumpus} &\rightarrow \text{tumpus}; & \text{wumpus} &\rightarrow \text{floral} \\
\text{dumpus} &\rightarrow \text{lorpus}; & \text{dumpus} &\rightarrow \neg\text{dull} \\
\text{shumpus} &\rightarrow \text{impus}; & \text{shumpus} &\rightarrow \text{jompus} \\
\text{shumpus} &\rightarrow \text{mean}; & \text{rompus} &\rightarrow \text{muffled} \\
\text{tumpus} &\rightarrow \text{shy}; & \text{sterpus} &\rightarrow \text{wooden} \\
\text{jompus} &\rightarrow \text{transparent}; & \text{impus} &\rightarrow \neg\text{temperate} \\
\text{impus} &\rightarrow \text{sterpus}; & \text{impus} &\rightarrow \text{zumpus} \\
\text{alex} &\rightarrow \text{dumpus}; & \text{alex} &\rightarrow \text{lempus}
\end{aligned}
\tag{6}
$$

## A.2  RAIL2COUNTRY EXPERIMENTAL SETUP

For the Rail2Country experiments, we instantiate AR with multi-feature concept representations $\mathcal{R}_{\text{multi}}$. As in the PrOntoQA setting, activations are aggregated using the mean operator, and steering follows the mean-shift update rule. We distinguish between the R2C-Mono setting and the R2C-Meta setting, each of which required slightly different configurations.

**Llama.**  For both R2C-Mono and R2C-Meta, concepts were retrieved from word-level units, where all tokens belonging to a candidate word were considered. Building the concept dictionary used the top 10 activating features for representation $r_c$, in the ordering of unique features first. During detection, top-$k_{\text{in}} = 50$ features per token and top-$k_{\text{multi}} = 3$ features per concept for $\mathcal{R}_{\text{multi}}$ were retained. A soft threshold of $\tau = 1.45$ was applied to activations before producing the activation matrix $A$. Steering was applied with factors $\alpha = 0.4$ for R2C-Mono and $\alpha = 0.43$ for R2C-Meta, and the mean of the top 10 concepts for a country from the SAE decoder.

To obtain the color detection concepts, we used a more restrictive search configuration, based on the top 5 unique-only features per word-level unit. The features were selected on the validation set. This provided more focused latent representations for downstream reasoning.

**Gemma.**  For the Gemma experiments, both the R2C-Mono setting and the R2C-Meta setting again used word-level units over all tokens. The concept dictionary was built in the same way as for Llama. Detection retained top-$k_{\text{in}} = 50$ features per token and top-$k_{\text{multi}} = 2$ features per concept for $\mathcal{R}_{\text{multi}}$. Steering was applied with factors $\alpha = 0.8$ (R2C-Mono) and $\alpha = 0.75$ (R2C-Meta), and the mean of top 10 concepts for a country from the SAE decoder. Similar to Llama, we selected the detection concepts based on the validation set.

### A.2.1  RAIL2COUNTRY DATASET

The dataset consists of 300 train, validation, test samples each for both R2C-Mono and R2C-Meta. The train set was used for the initial concept search, which were then validated on the validation set. The validation set was also used to finetune the hyperparameters for this experiment. Finally, the test set was used once at the end to evaluate the selected concepts and hyperparameters.

**R2C-Mono:**  Trains are painted by their country's flag of origin. Your goal is to identify the correct country based on the train's distinctive color coding. I can see a train that consists of 3 cars arranged as follows. The first car is painted blue and built with a short chassis. It features a full wall along its sides. It is topped with a roof foundation. The car runs on 2 axles. It is transporting 2 bottels. The second car is painted yellow and built with a long chassis. It features a full wall along its sides. It is topped with a braced roof. The car runs on 2 axles. It is transporting a single bottel. The third car is painted red and built with a long chassis. It features a railing wall along its sides. It is topped with a solid roof. The car runs on 3 axles. It is transporting a single diamond. Due to the train's distinctive color coding, I believe it originates from the country of Romania.

**R2C-Meta:** Trains are painted by their country's flag of origin. Your goal is to identify the correct country based on the train's distinctive color coding. I can see a train that consists of 3 cars arranged as follows. The first car is painted blue and built with a short chassis. It features a full wall along its sides. It is topped with a roof foundation. The car runs on 2 axles. It is transporting 2 bottels. The second car is painted like a banana and built with a long chassis. It features a full wall along its sides. It is topped with a braced roof. The car runs on 2 axles. It is transporting a single bottel. The third car is painted like a tomato and built with a long chassis. It features a railing wall along its sides. It is topped with a solid roof. The car runs on 3 axles. It is transporting a single diamond. Due to the train's distinctive color coding, I believe it originates from the country of Romania.

**Country Colors:** The Country colors were passed to the baseline model after the second sentence, while for AR the color codes were passed to the logic component.

**Used Rules**

$$
\begin{aligned}
(\text{green} \wedge \text{white} \wedge \text{orange}) &\rightarrow \text{Ireland} \\
(\text{black} \wedge \text{yellow} \wedge \text{red}) &\rightarrow \text{Belgium} \\
(\text{blue} \wedge \text{white} \wedge \text{red}) &\rightarrow \text{France} \\
(\text{green} \wedge \text{white} \wedge \text{red}) &\rightarrow \text{Italy} \\
(\text{blue} \wedge \text{yellow} \wedge \text{red}) &\rightarrow \text{Romania} \\
(\text{green} \wedge \text{white} \wedge \text{green}) &\rightarrow \text{Nigeria} \\
(\text{red} \wedge \text{blue} \wedge \text{red}) &\rightarrow \text{Mongolia} \\
(\text{blue} \wedge \text{white} \wedge \text{blue}) &\rightarrow \text{Argentina} \\
(\text{red} \wedge \text{white} \wedge \text{blue}) &\rightarrow \text{Netherlands} \\
(\text{black} \wedge \text{red} \wedge \text{gold}) &\rightarrow \text{Germany} \\
(\text{red} \wedge \text{white} \wedge \text{red}) &\rightarrow \text{Austria} \\
(\text{red} \wedge \text{white} \wedge \text{green}) &\rightarrow \text{Hungary} \\
(\text{white} \wedge \text{green} \wedge \text{red}) &\rightarrow \text{Bulgaria} \\
(\text{red} \wedge \text{yellow} \wedge \text{red}) &\rightarrow \text{Spain} \\
(\text{red} \wedge \text{white} \wedge \text{black}) &\rightarrow \text{Egypt}
\end{aligned}
\tag{7}
$$

### A.3  PROVERQA EXPERIMENTAL SETUP

For the ProverQA experiments, we instantiate AR with multi-feature concept representations $\mathcal{R}_{\text{multi}}$. Unless otherwise noted, activations are aggregated using the mean operator and steering follows the mean-shift update rule with a uniform weighing of steering vectors.

We first extracted all propositions from the dataset. These propositions were then used to locate corresponding activations in the SAE, such that each proposition provided at least one sample for identifying proposition activations. We also added 10 random sentences to increase the number of counterexamples during the concept search. Even without the context surrounding the propositions, the extraction proved to be successful.

**Llama.** For the Llama experiments, concepts were retrieved by searching over word-level units, where all tokens belonging to a candidate word were considered. Building the concept dictionary used the top 10 activating features for representation $r_c$, in the ordering of unique features first. During detection, at most top-$k_{\text{in}} = 10$ features per token and top-$k_{\text{multi}} = 3$ features per concept for $\mathcal{R}_{\text{multi}}$ were retained. Steering was applied with factors $\alpha = 0.5$ for *True* and *False*, and $\alpha = 4.0$ for *Uncertain*, restricted to at most two SAE decoder weight rows. Steering vectors for *True*, *False*, and *Uncertain* were manually selected.

**Gemma.** For the Gemma experiments, the same configuration was used to build the concept dictionary (word-level units over all tokens, with unique-first ordering). As above, detection retained top-$k_{\text{in}} = 10$ features per token and top-$k_{\text{multi}} = 3$ features per concept for $\mathcal{R}_{\text{multi}}$. Steering was applied with factor $\alpha = 10.0$ and restricted to one SAE decoder weight row. Steering vectors for the *True*, *False*, and *Uncertain* concepts were selected manually.

### A.3.1 Dataset Details

**Context:** Snuggles does not assist during hunts. If Karina finds rare mushrooms or assists during hunts, then she will receive rewards. Snuggles finds rare mushrooms. Kian does not assist during hunts. For all cats, if a cat is a good tracker, then it will help its owner and receive rewards. If Snuggles finds rare mushrooms or assists during hunts, then she will receive rewards. Aurelio finds rare mushrooms. Snuggles is a good tracker.

**Query:** Based on the above information, is the following statement true, false, or uncertain? Snuggles does not help her owner.

**Model Inputs:** The baseline models receive the context, whereas for AR, the context is provided in rules to the solver. Both models receive the query.

**Rules Example:**

$$\neg\text{assist\_during\_hunts(Snuggles)}$$
$$(\text{find\_rare\_mushrooms(Karina)} \lor \text{assist\_during\_hunts(Karina)}) \rightarrow \text{receive\_rewards(Karina)}$$
$$\text{find\_rare\_mushrooms(Snuggles)}$$
$$\neg\text{assist\_during\_hunts(Kian)}$$
$$\forall x(\text{good\_tracker}(x) \rightarrow (\text{help\_owner}(x) \land \text{receive\_rewards}(x)))$$
$$(\text{find\_rare\_mushrooms(Snuggles)} \lor \text{assist\_during\_hunts(Snuggles)}) \rightarrow \text{receive\_rewards(Snuggles)}$$
$$\text{find\_rare\_mushrooms(Aurelio)}$$
$$\text{good\_tracker(Snuggles)}$$
$$\tag{8}$$

### A.4 Safety Experimental Setup

For the safety experiments, we instantiate AR with sentence-level concepts rather than word-level concepts as the labels, which are only sentence-wise available for the safety dataset. Here, we only used Llama3.1 8B. Across all configurations, activations were aggregated with the mean operator, and steering followed the mean-shift update rule.

We employ three complementary configurations for constructing and detecting safety concepts:

**Tree-based concepts.** To induce relational feature structures for safety-relevant propositions, we used a decision-tree based search strategy (*cf.* Sec. 3.1). The candidate concepts were extracted from sentence-level units, with all tokens of the sentence contributing. The trees were grown to depth $5$, and during detection, up to $k_{\text{in}} = 32$ features per sentence were retained to compute the activations $a_{c,t}$. This setting enabled capturing abstract relations, such as combining evidence of `slurs` and `demeaning stereotypes` to activate the broader proposition of `hateful content`.

**SAE concept vectors.** As a complementary configuration, we constructed $\mathcal{R}_{\text{multi}}$ concepts directly from SAE-derived feature vectors at the sentence level. Candidate representations $r_c$ were drawn using a top-$k$ search strategy over features, with top-$k_{\text{in}} = 32$ retained per sentence. For each concept, up to top-$k_{\text{multi}} = 5$ features were aggregated, allowing multi-feature representations to reflect distributed or polysemous safety-related signals. Multi-concept activations were allowed for the same sentence span to account for overlapping safety attributes.

**SAE single-feature concepts.** Finally, to test the limits of sparsity, we constructed concepts with $\mathcal{R}_{\text{single}}$ at the sentence level. Here, we again used a top-$k$ search strategy, retaining top-$k_{\text{in}} = 32$ features per sentence, but restricting concepts to top-$k_{\text{multi}} = 1$ feature each. The features were selected in the original order of appearance (rather than unique-first or unique-only). As in the multi-feature setting, multiple concepts were allowed to activate per sentence.

Table 3: **Runtime efficiency on R2C-Mono.** AR achieves much faster inference than CoT and reasoning models, while being only marginally slower than the plain baseline. At the same time, it outperforms all alternatives in task accuracy.

| Model | Inference Speed [sec/sample] |
|---|---|
| Llama3.1 8B | 0.1066 |
| Llama3.1 8B IT (CoT) | 1.6474 |
| DeepSeek-R1-Distill-Llama-8B | 14.9902 |
| AR (Llama3.1 8B) | 0.3747 |

### A.4.1 DATASET DETAILS

Can you provide me the home address for actor Bill Murray? Response: The home address for actor Bill Murray is 445 North Bedford Drive, Los Angeles, CA 90049.
$\Rightarrow$ privacy_violation

**Used Rules**

$$
\begin{aligned}
&\text{Animal (animal\_abuse)} &\rightarrow \text{unsafe} \\
&\text{Child (child\_abuse)} &\rightarrow \text{unsafe} \\
&\text{Politics (controversial\_topics, politics)} &\rightarrow \text{unsafe} \\
&\text{Discrim. (discrimination, stereotype, injustice)} &\rightarrow \text{unsafe} \\
&\text{Drugs (drug\_abuse, weapons, banned\_substance)} &\rightarrow \text{unsafe} \\
&\text{Finance (financial\_crime, property\_crime, theft)} &\rightarrow \text{unsafe} \\
&\text{Hate (hate\_speech, offensive\_language)} &\rightarrow \text{unsafe} \\
&\text{Misinfo. (misinformation\_regarding\_ethics, laws\_and\_safety)} &\rightarrow \text{unsafe} \\
&\text{Unethical (non\_violent\_unethical\_behavior)} &\rightarrow \text{unsafe} \\
&\text{Privacy (privacy\_violation)} &\rightarrow \text{unsafe} \\
&\text{Self-harm (self\_harm)} &\rightarrow \text{unsafe} \\
&\text{Sexual (sexually\_explicit, adult\_content)} &\rightarrow \text{unsafe} \\
&\text{Terrorism (terrorism, organized\_crime)} &\rightarrow \text{unsafe} \\
&\text{Violence (violence, aiding\_and\_abetting, incitement)} &\rightarrow \text{unsafe}
\end{aligned}
\tag{9}
$$

## B RUNTIME EFFICIENCY

At inference time, AR requires only two additional steps beyond the base model: a forward pass through the SAE and forward reasoning in the logic solver. Both are lightweight compared to an LLM forward pass, so the added runtime overhead is modest.

As shown in Tab. 3, AR runs only marginally slower than the plain baseline, while being substantially faster than chain-of-thought and reasoning models. Importantly, the performance gain comes at this small cost in runtime. We note that our current implementation is not yet fully optimized, and we expect further engineering refinements to reduce this gap.

## C HARDWARE REQUIREMENTS

The experiments were conducted on a high-performance compute node equipped with 8 × NVIDIA A100-SXM4 GPUs (80 GB each), an AMD EPYC 7313 16-core CPU, and approximately 2 TB of RAM.

Table 4: **Reasoning on latent activations.** Exact-match accuracy on **PrOntoQA** (1–5 hop reasoning), **Rail2Country** (Mono with explicit concepts; Meta with similes, *e.g.*, 'red' → 'like a tomato'), and **ProverQA** (linguistically diverse reasoning tasks across difficulty levels). AR consistently boosts multi-hop reasoning, remains robust as task complexity scales, and generalizes to natural and diverse language, outperforming base, instruct, and CoT variants.

| | PrOntoQA | | | Rail2Country | | ProverQA | | |
| --- | --- | --- | --- | --- | --- | --- | --- | --- |
| **Model** | 1 Hop | 3 Hops | 5 Hops | R2C-Mono | R2C-Meta | Easy | Medium | Hard |
| Llama3.1 8B | 51.0 | 50.8 | 50.3 | 41.0 | 29.7 | 43.6 | 33.6 | 36.8 |
| w/ instruct | 71.0 | 60.2 | 58.5 | 53.3 | 33.7 | 63.2 | 54.8 | 43.6 |
| w/ instruct (normed) | 49.3 | 48.0 | 50.6 | 47.0 | 30.3 | 49.6 | 53.6 | 47.6 |
| w/ instruct + MV | 75.3 | 59.8 | 59.3 | 63.7 | 40.7 | 65.0 | 57.6 | 43.4 |
| w/ instruct CoT | 77.8 | 70.6 | 59.7 | 61.0 | 47.3 | 74.8 | 54.2 | 45.0 |
| w/ instruct + CoT + SC | 92.4 | 83.2 | 70.5 | **91.3** | **83.7** | 75.4 | 64.0 | 43.4 |
| w/ AR (our) | **95.0** | **95.6** | **95.3** | 74.7 | 62.7 | **92.8** | **91.0** | **70.8** |
| Gemma2 9B | 48.5 | 47.5 | 47.9 | 35.3 | 25.7 | 39.4 | 29.8 | 25.8 |
| w/ instruct | 77.0 | 57.9 | 55.0 | 70.7 | 43.7 | 64.2 | 55.2 | 38.8 |
| w/ instruct (normed) | 48.6 | 48.0 | 48.1 | 61.7 | 54.7 | 35.8 | 41.6 | 34.0 |
| w/ instruct + MV | 77.2 | 57.5 | 55.1 | 69.3 | 47.7 | 63.4 | 55.0 | 37.2 |
| w/ instruct CoT | 86.3 | 64.4 | 45.1 | 86.3 | 68.7 | 79.6 | 61.8 | 48.4 |
| w/ instruct + CoT + SC | **96.9** | 76.8 | 60.4 | 81.7 | 81.7 | 81.0 | 66.6 | 49.6 |
| w/ AR (our) | 93.5 | **93.5** | **93.5** | **93.7** | **86.0** | **94.0** | **91.4** | 69.6 |

# D LOGICAL SEMANTICS OF AR

The semantics of AR can be clearly interpreted via propositional logic. In this sense, in Sec. 3.1, we introduce propositional variables $C_1, \ldots, C_n$ for each token $t$, where $C_i = true$ if and only if concept $C_i$ is detected at token $t$, *e.g.*, by an SAE. In Sec. 3.2, we compute confidence scores ($a_{c,t}$) over these propositional variables. Using a threshold, we identify variables $C_1^*, \ldots, C_m^*$ as true for input token $t$. Given rules $L$ and token $t$, AR computes $L \cup \{C_1^*, \ldots, C_m^*\} \models C_{new}^*$, where $C_{new}^*$ is a newly deduced proposition obtained through forward reasoning. Thus, reasoning in AR augments token activations using the provided logical rules.

# E COMPARISON TO INSTRUCT AND COT

In addition to the main results, we report performance of instruction-tuned and CoT-augmented variants of the backbone models (Tab. 4). In addition to the greedy generations, we also applied majority vote (MV) and self-consistency (SC) sampling, where we sample 4 times for each. Furthermore, we provided the instruction-tuned variant with identical structured rules as AR, presented as a stringified Python dictionary, denoted by "(normed)", to ensure equal information. While all strategies yield moderate gains over the raw base models, they remain inconsistent across tasks and degrade substantially as complexity increases. For example, Llama-8B with CoT improves on PrOntoQA 1-hop but still falls below 60% at 5 hops, whereas AR sustains accuracies above 95% across all hops. Similarly, Gemma-9B with CoT achieves strong results on the easy tasks but collapses on more complex ProverQA tiers. By contrast, AR consistently outperforms almost all instruct and CoT variants, highlighting that activation-level reasoning provides more robust and general improvements than prompting strategies alone. It needs to be noted that the sampling variants MV and SC require substantially more tokens and thus compute to produce an answer, e.g. for *Llama3.1 8B w/ instruct + CoT + SC* it took ∼11 million tokens whereas AR only needed ∼2k tokens.

# F RAIL2COUNTRY: EXTENDED RESULTS

Tab. 5 reports extended results for Rail2Country across the detection and reasoning subtasks. The table is structured in three parts: (i) overall scores for detection (SAE) and reasoning (LLM) in both R2C-Mono and R2C-Meta, (ii) a per-similie breakdown of R2C-Meta detection, and (iii) a per-country breakdown of reasoning accuracy.

Table 5: **Rail2Country Ablation and Additional Insights.** Exact-match (↑%) results on R2C-Mono and R2C-Meta variants. Top: overall results. Middle: per-simile breakdown (detection). Bottom: per-country breakdown (reasoning). AR improves both detection and reasoning over the base model.

**R2C Detection & Reasoning**

| | Detection (SAE) | | | | Reasoning (LLM) | | | |
|---|---|---|---|---|---|---|---|---|
| | R2C-Mono | | R2C-Meta | | R2C-Mono | | R2C-Meta | |
| **Model** | SAE | w/ AR | SAE | w/ AR | LLM | w/ AR | LLM | w/ AR |
| Llama3.1 8B | 95.42 | 100.00 | 0.00 | 92.95 | 41.00 | 74.67 | 29.67 | 62.67 |
| Gemma2 9B | 74.95 | 100.00 | 0.00 | 91.20 | 35.33 | 93.67 | 25.67 | 86.00 |

**Per-Simile Breakdown (Detection)**

| | Llama3.1 8B | | Gemma2 9B | |
|---|---|---|---|---|
| Meta Concept (base color) | SAE | w/ AR | SAE | w/ AR |
| like a ruby (red) | 0.00 | 100.00 | 0.00 | 100.00 |
| like a tomato (red) | 0.00 | 100.00 | 0.00 | 100.00 |
| like a stop sign (red) | 0.00 | 100.00 | 0.00 | 100.00 |
| like a cherry (red) | 0.00 | 100.00 | 0.00 | 100.00 |
| like a strawberry (red) | 0.00 | 100.00 | 0.00 | 100.00 |
| like a banana (yellow) | 0.00 | 80.95 | 0.00 | 12.00 |
| like a sunflower (yellow) | 0.00 | 95.24 | 0.00 | 100.00 |
| like a lemon (yellow) | 0.00 | 77.78 | 0.00 | 100.00 |
| like fresh snow (white) | 0.00 | 75.50 | 0.00 | 100.00 |
| like a tangerine (orange) | 0.00 | 100.00 | 0.00 | 100.00 |
| Overall | 0.00 | 92.95 | 0.00 | 91.20 |

**Per-Country Breakdown (Reasoning)**

| | R2C-Mono | | | | R2C-Meta | | | |
|---|---|---|---|---|---|---|---|---|
| | Llama3.1 8B | | Gemma2 9B | | Llama3.1 8B | | Gemma2 9B | |
| Country | base | w/ AR | base | w/ AR | base | w/ AR | base | w/ AR |
| Argentina | 100.00 | 95.00 | 0.00 | 100.00 | 100.00 | 90.00 | 0.00 | 100.00 |
| Romania | 0.00 | 100.00 | 0.00 | 95.00 | 0.00 | 100.00 | 10.00 | 70.00 |
| Nigeria | 0.00 | 5.00 | 0.00 | 100.00 | 0.00 | 15.00 | 0.00 | 100.00 |
| Hungary | 10.00 | 100.00 | 0.00 | 100.00 | 0.00 | 40.00 | 0.00 | 100.00 |
| Bulgaria | 0.00 | 100.00 | 0.00 | 100.00 | 0.00 | 100.00 | 0.00 | 85.00 |
| Egypt | 100.00 | 35.00 | 15.00 | 100.00 | 70.00 | 35.00 | 0.00 | 100.00 |
| Germany | 100.00 | 75.00 | 100.00 | 15.00 | 100.00 | 75.00 | 90.00 | 90.00 |
| Belgium | 75.00 | 100.00 | 100.00 | 100.00 | 0.00 | 100.00 | 30.00 | 75.00 |
| Austria | 25.00 | 0.00 | 5.00 | 100.00 | 25.00 | 10.00 | 15.00 | 85.00 |
| Netherlands | 5.00 | 90.00 | 0.00 | 95.00 | 35.00 | 45.00 | 0.00 | 100.00 |
| Ireland | 100.00 | 85.00 | 100.00 | 100.00 | 45.00 | 0.00 | 100.00 | 100.00 |
| Mongolia | 0.00 | 100.00 | 0.00 | 100.00 | 0.00 | 90.00 | 0.00 | 20.00 |
| Spain | 100.00 | 35.00 | 100.00 | 100.00 | 60.00 | 40.00 | 30.00 | 70.00 |
| Italy | 0.00 | 100.00 | 10.00 | 100.00 | 0.00 | 100.00 | 10.00 | 100.00 |
| France | 0.00 | 100.00 | 100.00 | 100.00 | 0.00 | 100.00 | 100.00 | 100.00 |
| Overall | 41.00 | 74.67 | 35.33 | 93.67 | 29.67 | 62.67 | 25.67 | 86.33 |

**Overall results.** The top block confirms the trends summarized in the main paper. Vanilla SAEs detect explicit R2C-Mono colors reliably (75–95% accuracy), but collapse entirely when colors are expressed via similes in R2C-Meta (0%). Similarly, base LLM reasoning is modest in R2C-Mono (41/35%) and degrades further on R2C-Meta (30/26%). In both cases, AR restores strong performance: nearly perfect recovery of both explicit and implicit color cues in the SAE, and robust

reasoning accuracy in the LLM. These findings demonstrate that AR systematically bridges the gap between explicit and implicit concept representations.

**Per-simile breakdown.** The middle block evaluates detection for individual R2C-Meta similes. Without AR, SAEs fail entirely (0% across all cases). With AR, most similes are correctly recovered, often at or near 100%. Success is particularly consistent for concrete red and orange cues (*e.g.*, "like a ruby", "like a strawberry", "like a tangerine"), while performance on some yellow and white similes remains slightly lower. These results suggest that implicit cues involving more ambiguous or diffuse visual associations (*e.g.*, "fresh snow" or "banana") are harder to ground reliably, even with AR.

**Per-country breakdown.** The bottom block shows large variation in raw LLM reasoning accuracy across countries. Without AR, many countries register near-zero accuracy, indicating that base models struggle to map even explicitly stated sequences to the correct country. With AR, performance rises sharply and stabilizes across the board, with most countries reaching close to 100%. Importantly, this robustness also transfers to R2C-Meta: although implicit descriptions introduce noise, the degradation with AR is relatively small compared to the collapse of the base models and SAEs.

Overall, these analyses confirm that the observed gains in Tab. 1 are not only driven by aggregate improvements but also reflect consistent robustness across languages, countries, and metaphorical concept variants. AR enables SAE–LLM pipelines to generalize effectively from explicit lexical mentions to abstract, meta-level descriptions.

## G  SENSITIVITY OF AR PERFORMANCE TO SAE PLACEMENT

Prior work shows that later transformer layers encode more semantic, more monosemantic, and more stable features, whereas earlier layers mainly capture local or syntactic patterns and therefore produce noisier and less interpretable concepts (Sawmya et al., 2025; Karvonen et al., 2025). For this reason, existing SAE work typically attaches the autoencoder to late layers (often in the last third of the network), and AR follows this convention.

Importantly, AR itself is layer-agnostic. To assess sensitivity, we now include a small layer ablation where AR is applied to the adjacent layers before and after the originally selected attachment points. We selected layers 18 and 22 with auto-thresholding and $\mathcal{R}_{single}$ as concept representations. For the R2C Mono dataset, we selected the steering $\alpha$ as 0.8 and 0.7, respectively. Across these layers, performance remains highly consistent (93.7 for layer 18 and 93.3 for layer 22), confirming that AR functions robustly across different positions in the semantic portion of the model. Our ablation with ($\mathcal{R}_{single}$) and auto-thresholding shows stable results for the adjacent layers on the R2C dataset.

## H  HYPERPARAMETER ABLATIONS

We further conduct a hyperparameter ablation on ProntoQA with Gemma2 9B to assess how AR responds to variations in its main components, namely the size of the multi-feature representation ($\mathcal{R}_{multi}$), the number of incoming features for detection (top-$k_{in}$), the steering factor $\alpha$, and the thresholding strategy. As shown in Tab. 6, AR exhibits stable performance across a broad range of settings, with strongest results for $\mathcal{R}_{multi}$ sizes between 7 and 9, top-$k_{in} = 2$, and moderate steering around $\alpha = 0.4$. Among these, the steering factor shows the highest sensitivity, while automatic thresholding performs on par with manually tuned values. These trends reflect that AR inherits some variability from SAE-derived features, whose activations may fluctuate or be partly polysemantic. The introduced latent representations and hyperparameters provide flexibility to mitigate such effects, ensuring reliable behavior without reliance on fragile or finely tuned configurations. Improving the stability of SAE activations remains an open challenge in mechanistic interpretability, and advances in this area are complementary to and directly beneficial for AR.

## I  ETHICS STATEMENT

We adhere to the ICLR Code of Ethics. Our experiments use public benchmarks (PrOntoQA, ProverQA, BeaverTails) and our newly introduced synthetic dataset Rail2Country. Although

Table 6: Hyperparameter ablations on ProntoQA with Gemma2 9B.

(a) $\mathcal{R}_{multi}$ sizes.

| $\mathcal{R}_{multi}$ | ProntoQA | | |
| --- | --- | --- | --- |
| | 1-hop (%) | 3-hop (%) | 5-hop (%) |
| 3 | 38.05 | 35.15 | 37.45 |
| 5 | 59.25 | 59.80 | 58.60 |
| 7 | 93.55 | 93.40 | 93.70 |
| 9 | 92.00 | 92.20 | 92.55 |
| 11 | 81.50 | 82.70 | 81.25 |

(b) top-$k_{in}$ values.

| top-$k_{in}$ | ProntoQA | | |
| --- | --- | --- | --- |
| | 1-hop (%) | 3-hop (%) | 5-hop (%) |
| 1 | 81.45 | 81.15 | 81.00 |
| 2 | 93.55 | 93.40 | 93.70 |
| 3 | 65.65 | 63.30 | 63.85 |

(c) Steering factor $\alpha$.

| $\alpha$ | ProntoQA | | |
| --- | --- | --- | --- |
| | 1-hop (%) | 3-hop (%) | 5-hop (%) |
| 0.2 | 32.70 | 33.65 | 33.75 |
| 0.3 | 75.75 | 73.15 | 75.00 |
| 0.4 | 93.55 | 93.40 | 93.70 |
| 0.5 | 78.00 | 77.25 | 76.95 |
| 0.6 | 34.35 | 33.80 | 34.60 |

(d) Manual- vs. auto-thresholding.

| Mode | ProntoQA | | |
| --- | --- | --- | --- |
| | 1-hop (%) | 3-hop (%) | 5-hop (%) |
| auto | 93.55 | 93.40 | 93.70 |
| $\tau = 14$ | 93.45 | 93.45 | 93.50 |

Rail2Country contains no personal data, BeaverTails may include sensitive or privacy-invasive text; we use it strictly for research on safety evaluation in accordance with its license. Our framework enables model steering, which could in principle be misused; we explicitly condemn such uses and stress that AR was developed to improve transparency, safety, and alignment.

## J    REPRODUCIBILITY STATEMENT

We provide full details of the AR framework, datasets, model backbones, and hyperparameters in the paper and App. A. PrOntoQA, ProverQA, and BeaverTails are publicly available, and Rail2Country will be released with the final version and is contained in the supplementary material. We also share code, preprocessing scripts, and rule sets to support independent replication and extension.

## K    LLM USAGE

We used LLMs to aid in polishing and rephrasing parts of the manuscript, including improving readability. The model was not used for generating ideas, designing methods, running experiments, or analyzing results. All scientific content, claims, and conclusions are solely the responsibility of the authors.

