# OpenReview forum: "ActivationReasoning: Logical Reasoning in Latent Activation Spaces"
_ICLR.cc/2026/Conference — ICLR 2026 Poster_

### Official Review · Reviewer_tgpL · 2025-10-21

**Soundness:** 2
**Presentation:** 3
**Contribution:** 3
**Rating:** 4
**Confidence:** 2

**Summary:**

The paper proposes ACTIVATIONREASONING (AR), a framework that maps sparse autoencoder (SAE) features in LLMs to logical propositions and performs forward-chaining deduction over them to compose higher-level concepts and steer generation. Pipeline: (1) build a concept dictionary from SAE features (single/multi/relational representations), (2) detect token/sequence-level activations with soft thresholds, (3) apply logic rules to infer new propositions (A → A′) and optionally steer model activations. Experiments on PrOntoQA, a new Rail2Country benchmark (explicit vs. simile color cues), ProverQA, and BeaverTails safety show large gains over base and some larger/“reasoning” models, reportedly maintaining accuracy as reasoning depth increases and improving safety detection; authors claim small runtime overhead.

**Strengths:**

S1. Modular pipeline: concept discovery (via SAEs), activation aggregation, logic layer, and steering—each replaceable/ablatable.

S2. Strong multi-hop robustness; recovery of implicit/meta cues (similes); safety composition via logic; low runtime overhead vs. CoT/reasoning models.

**Weaknesses:**

W1.  The method is fundamentally deductive: it thresholds SAE features into propositions and applies forward-chaining rules. But domain-general reasoning (esp. scientific reasoning) is largely inductive/abductive—forming and revising hypotheses under uncertainty, trading off competing explanations, and learning new rules from sparse evidence. Here, rules are user-provided or templated; there’s no rule induction, hypothesis revision, or probabilistic belief update. The evaluations don’t demonstrate transfer to open-class concepts, concept drift, or out-of-ontology settings where the correct rule schema is unknown. As a result, the approach risks brittleness outside closed-world toy tasks: once a needed predicate or relation is missing (or mis-grounded by the SAE), the system cannot generalize by learning—it can only fail silently or require manual rule engineering. For a claim of “domain-general” promise, I’d expect ablations with (i) novel domains without preencoded rules, (ii) noisy/contradictory evidence with uncertainty propagation, and (iii) an inductive component (e.g., differentiable rule learning, Bayesian updates, or abductive reasoning) that improves with more data rather than more hand-written logic.

W2.  The paper compares AR to CoT and “reasoning models” but runs everyone with greedy decoding and without widely-accepted best practices (e.g., self-consistency, majority voting, verifier reranking, program-of-thought, retrieval-augmented logic parsers). A fair test would include: (i) symbolic parsers + classical prover, (ii) pure-symbolic systems on the same rule input AR gets, (iii) LLMs given the same normalized rule form.

W3.  Multiple places admit selecting features/hyperparameters on evaluation data (e.g., Rail2Country: “features were selected on the evaluation set” and thresholds tuned on eval/test splits). This is classic double-dipping that inflates reported gains. A rigorous protocol would fix search on train only, lock thresholds, and report test once.

**Questions:**

Can you freeze a single, pre-registered recipe (feature search, thresholds τ, top-k, α, steering vectors) learned on train only, and report test once with CIs across seeds? Provide ablations showing performance sensitivity to each knob.

---

> ### Author Response · Authors · 2025-11-19
>
> We thank reviewer tgpL for the thorough review and the broad set of questions raised. These touch on several interesting aspects of our approach, and we are grateful for the opportunity to further expand and clarify our methodological choices. We respond to each point in detail below.
>
> ## W1 (inductive/abductive reasoning and beyond closed-world, rule-engineered tasks)
> Indeed, the current implementation of ActivationReasoning (AR) focuses on deductive reasoning via forward chaining. This design choice was deliberate, since our primary goal is to show that it is feasable to perform logical reasoning within latent activation spaces of LLMs in a transparent and controlable way. That said, AR’s architecture is agnostic to the reasoning paradigm. The deductive solver can be seamlessly replaced by inductive or abductive engines such as Aleph [1] or Popper [2], or by probabilistic solvers like SLIPCOVER [3] or ProbFOIL [4], which naturally support hypothesis induction and uncertainty propagation. We will expand the limitations section to highlight this as a natural extension for future work.
>
> We disagree regarding "brittleness outside closed-world toy tasks". Our current benchmarks already go beyond simple toy domains. ProverQA is an open-world logic benchmark that makes use of unconstrained natural-language inputs. Further, BeaverTails evaluates real-world safety reasoning tasks involving abstract, non-symbolic concepts. These results highlight AR’s capacity to generalize beyond closed-world toy settings. However, we do agree that mis-grounded SAE features are a hot topic in many SAE papers [5,6,7,8] and remain an orthogonal challenge also for AR's current instantiation.
>
> Further, we agree that broader evaluations involving rule induction or explicit uncertainty propagation can potentially extend the scope of the work. However, these are orthogonal to our main contribution and fall outside the scope of what we aim to address in this paper. We are thankful for the suggestions and will adjust the framing and discussion to reflect this more clearly.
>
> [1] Aleph (Srinivasan, 2001)
>
> [2] Popper (Cropper & Morel, 2021)
>
> [3] SLIPCOVER (Bellodi & Riguzzi, Theory and Practice of Logic Programming 15.2, 2015)
>
> [4] ProbFOIL (De Raedt et al., IJCAI, 2015)
>
> [5] Measuring and Guiding Monosemanticity (Härle et al., NeurIPS 2025)
>
> [6] Learning Multi-Level Features with Matryoshka Sparse Autoencoders (Bussmann et al., ICML 2025)
>
> [7] SAEBench: A Comprehensive Benchmark for Sparse Autoencoders in Language Model Interpretability (Karvonen et al., 2025)
>
> [8] Open Problems in Mechanistic Interpretability (Sharkey et al., TMLR 2025)

---

> > ### Author Response · Authors · 2025-11-19
> >
> > ## W2 (baseline comparison; LLM reasoning practices and symbolic baselines)
> > All LLM baselines in the submission were deliberately evaluated under identical conditions using greedy decoding to avoid stochastic variance and isolate the reasoning effect. Results with no contraints via GPT-4o were in fact obtained via sampling (due to API constraints), which AR still outperforms.
> > Nevertheless, we have expanded our evaluation via Llama3.1 and Gemma2 to include other widely used decoding strategies such as majority voting (mv) and self-consistency (sc), as well as an “LLM + normalized rules” condition (normed). These results (cf. tables below) will be included in the revised manuscript. We observe across all datasets that AR continues to outperform these settings by a substantial margin. Generally, self-consistency improves over plain CoT on simpler instances but degrades on harder multi-hop tasks and incurs very high inference costs (millions of tokens per benchmark), making it impractical for scaling.
> >
> > To be sure we are on the same page, regarding “symbolic parser + classical prover”: we understand a pipeline in which (i) natural-language inputs are first mapped into logical predicates by a dedicated parser (e.g., an LLM or a rule-based semantic parser), and (ii) these predicates, together with the rules, are given to an off-the-shelf logic engine (e.g., a Prolog/Datalog/ASP solver) for exact reasoning. We agree that this is a valuable comparison. For synthetic datasets such as ProntoQA and ProverQA, we can indeed instantiate a pure-symbolic upper bound by running a classical solver directly on the gold rules/facts. However, for open-ended safety datasets such as BeaverTails and Rail2Country there is no available symbolic parser that can reliably map nuanced phenomena (e.g., implicit hate, veiled threats) into logic predicates, so a clean “parser + prover” baseline is not directly applicable. This is precisely the gap AR aims to address: it retains the strengths of symbolic reasoning while grounding concepts in the latent space of an LLM.
> >
> > We will integrate these clarifications and expanded baselines into the revised version.
> >
> > ### Additional Baselines
> > |mode|prontoqa 1-hop ACC|prontoqa 3-hop ACC|prontoqa 5-hop ACC|r2c mono ACC|r2c meta ACC|proverqa easy ACC|proverqa medium ACC|proverqa hard ACC|
> > |-|-|-|-|-|-|-|-|-|
> > |**Llama3.1 8B**|51.0%|50.8%|50.3%|41.0%|29.7%|43.6%|33.6%|36.8%|
> > |w/ it (normed)|49.3%|48.0%|50.6%|47.0%|30.3%|49.6%|53.6%|47.6%|
> > |w/ it + mv|75.3%|59.8%|59.3%|63.7%|40.7%|65.0%|57.6%|43.4%|
> > |w/ it + cot + sc|92.4%|83.2%|70.5%|91.3%|83.7%|75.4%|64.0%|43.4%|
> >
> > |mode|prontoqa 1-hop ACC|prontoqa 3-hop ACC|prontoqa 5-hop ACC|r2c mono ACC|r2c meta ACC|proverqa easy ACC|proverqa medium ACC|proverqa hard ACC|
> > |-|-|-|-|-|-|-|-|-|
> > |**Gemma2 9B**|48.5%|47.5%|47.9%|35.3%|25.7%|39.4%|29.8%|25.8%|
> > |w/ it (normed)|48.6%|48.0%|48.1%|61.7%|54.7%|35.8%|41.6%|34.0%|
> > |w/ it + mv|77.2%|57.5%|55.1%|69.3%|47.7%|63.4%|55.0%|37.2%|
> > |w/ it + cot + sc|96.9%|76.8%|60.4%|81.7%|81.7%|81.0%|66.6%|49.6%|

---

> > > ### Author Response · Authors · 2025-11-19
> > >
> > > ## W3+Q1 Train–Test Protocol and Hyperparameter Ablation
> > > We realize that our use of terminology in the Appendix may have been unclear, and we would like to clarify it here. The references to the “evaluation set” in the Appendix were intended to refer to the validation set. We will revise the section to clarify this. Thus, our workflow follows standard ML practice throughout, we use a strict train–validation–test protocol, and the test split is never accessed before final evaluation. Concretely, we proceed as follows: (i) Latent representations are obtained either through search on the training data, LLM-generated training data, or through manual assignment when SAE features are already well understood. (ii) Hyperparameters are selected using the validation split (or the training split if no validation split is provided). (iii) The test split is evaluated exactly once. The results reported in Tables 1 and 2 reflect this. Thus, all reported results use pre-registered hyperparameters selected exclusively from the train/validation split, and the **test set is only touched once**. We will make this procedure fully explicit in the revision.
> > >
> > > Further, we agree that ablating hyperparameter sensitivity is valuable. To this end, we have conducted an ablation study on ProntoQA (Gemma) to quantify how AR responds to variations in the latent representation size ($\mathcal R_{multi}$), steering factor ($\alpha$), detection granularity (top-$k_{in}$), and thresholding strategy. Overall (see Tables below), we find stable performance for $\mathcal R_{multi}=7-9, \alpha=0.4, top-k_{in}=2$, while the steering factor $\alpha$ exhibits the strongest sensitivity. In general, we recommend deriving thresholds automatically which we do, e.g., for our BeaverTails evaluations.
> > >
> > > Sensitivities stem from the fact that AR builds upon SAE-derived features, which can exhibit limitations such as polysemanticity and fluctuations in activation patterns. AR mitigates some of these effects through the introduced latent representations with additional flexibility provided via these hyperparameters, which allows users to control how latent activations are mapped to propositions and steering vectors.
> > >
> > > That said, improving the stability and reliability of SAE activations remains a relevant topic in ongoing research of mechanistic interpretability literature [5,6,7,8]. We believe that tackling this issue is beyond the primary scope of our work. Moreover, advances in SAE literature will naturally translate to increased robustness of AR, and we view our framework as complementary to these broader efforts.
> > >
> > > ## Ablation on different hyperparameter settings
> > > ### **1. $\mathcal R_{multi}$ sizes**
> > > |$\mathcal R_{multi}$ size|**1-hop (%)**|**3-hop (%)**|**5-hop (%)**|
> > > |-|-|-|-|
> > > |3|38.05|35.15|37.45|
> > > |5|59.25|59.80|58.60|
> > > |7|93.55|93.40|93.70|
> > > |9|92.00|92.20|92.55|
> > > |11|81.50|82.70|81.25|
> > >
> > > ### **2. top-$k_{in}$ values**
> > > |top-$k_{in}$|**1-hop (%)**|**3-hop (%)**|**5-hop (%)**|
> > > |-|-|-|-|
> > > |1|81.45|81.15|81.00|
> > > |2|93.55|93.40|93.70|
> > > |3|65.65|63.30|63.85|
> > >
> > > ### **3. Steering factor $\alpha$**
> > > |$\alpha$|**1-hop (%)**|**3-hop (%)**|**5-hop (%)**|
> > > |-|-|-|-|
> > > |0.2|32.70|33.65|33.75|
> > > |0.3|75.75|73.15|75.00|
> > > |0.4|93.55|93.40|93.70|
> > > |0.5|78.00|77.25|76.95|
> > > |0.6|34.35|33.80|34.60|
> > >
> > > ### **4. Activation Threshold**
> > > With thresholding vs. auto thresholding:
> > > |Hops|$\tau=14$|auto|
> > > |-|-|-|
> > > |1|93.45|93.55|
> > > |3|93.45|93.40|
> > > |5|93.50|93.70|
> > >
> > > ## References
> > > [1] Aleph (Srinivasan, 2001)
> > >
> > > [2] Popper (Cropper & Morel, 2021)
> > >
> > > [3] SLIPCOVER (Bellodi & Riguzzi, Theory and Practice of Logic Programming 15.2, 2015)
> > >
> > > [4] ProbFOIL (De Raedt et al., IJCAI, 2015)
> > >
> > > [5] Measuring and Guiding Monosemanticity (Härle et al., NeurIPS 2025)
> > >
> > > [6] Learning Multi-Level Features with Matryoshka Sparse Autoencoders (Bussmann et al., ICML 2025)
> > >
> > > [7] SAEBench: A Comprehensive Benchmark for Sparse Autoencoders in Language Model Interpretability (Karvonen et al., 2025)
> > >
> > > [8] Open Problems in Mechanistic Interpretability (Sharkey et al., TMLR 2025)

---

> ### Author Response · Authors · 2025-11-27
>
> As the discussion phase is drawing to a close, we would like to ask if there are any remaining concerns or clarifications you would like us to address. We would be glad to provide further details if helpful, and appreciate it if you would reconsider the scoring if no concerns remain. Thank you for your time and consideration.

---

### Official Review · Reviewer_aA2t · 2025-10-28

**Soundness:** 4
**Presentation:** 3
**Contribution:** 4
**Rating:** 6
**Confidence:** 2

**Summary:**

This paper proposes ACTIVATIONREASONING (AR), a novel framework for embedding **explicit logical reasoning into the latent activation space of LLMs. AR leverages SAEs  to extract monosemantic latent features that align with human-interpretable concepts and treats these features as **logical propositions**. The framework operates in three stages. This enables both transparent analysis of model internals and direct control over model behavior through concept-level interventions. Empirical evaluations on four benchmarks demonstrate that AR improves multi-hop reasoning, robustness to abstract or implicit cues, and context-sensitive safety control, often outperforming even much larger LLMs and reasoning-tuned models.

**Strengths:**

1. AR introduces a clear and principled framework that connects latent activations with symbolic logic, moving beyond typical mechanistic interpretability.

2. Results across diverse reasoning and safety tasks consistently show large gains, often surpassing significantly larger or reasoning-tuned baseline.

3. AR offers auditable reasoning and explicit model steering, addressing transparency and safety.

**Weaknesses:**

1. The proposed approach inherits limitations from sparse autoencoders. If latent features are polysemous or unstable, reasoning performance and interpretability may degrade.

2. It remains unclear how AR performs under high-dimensional activation spaces or longer contexts.

3. It would be valuable to discuss integration with probabilistic or continuous reasoning paradigms, which may align more naturally with neural representations.

**Questions:**

1. How scalable is AR when applied to large-scale tasks or long-context models, given the need to store and process activation matrices and logical rules?

2. Since the approach relies on SAEs trained independently from the LLM, how sensitive is AR’s performance to the specific choice or layer placement of the SAE?

3. Could the framework generalize to other modalities (e.g., vision–language models), where concept discovery and rule definition are more complex?

---

> ### Author Response · Authors · 2025-11-19
>
> We thank reviewer aA2t for their effort and thoughtful evaluation of our work. Several of the raised points touch on important aspects of our framework, and we address each of them in detail below.
>
> ## W1 (limitations inherited from sparse autoencoders due to polysemous or unstable latent features)
> We acknowledge and agree that SAEs can exhibit polysemous or unstable features, and AR can inherit these limitations. Yet, we wish to clarify that our primary contribution is not centered on improving SAEs themselves, but to show that once meaningful features are available, they allow for explicit logical reasoning in latent space. Importantly, AR is architecture-agnostic and is not constrained to a specific SAE architecture: any improved concept extractor (including more monosemantic SAEs as in [1]) can be plugged in without impacting the framework.
>
> At the same time, AR already mitigates several shortcomings of SAEs. First, it does not rely on monosemantic features: AR supports different representation types $\mathcal R_{single}$, $\mathcal R_{multi}$, $\mathcal R_{relation}$, which aggregate features and relations and thus reduces the impact of imperfect/individual polysemous units. Second, AR uses (auto-)thresholding to filter out low-activation noise, which helps isolate more reliable signals from imperfect SAE activations. As demonstrated in Table 2, the choice of representation can substantially improve both robustness and accuracy, even in the presence of imperfect activations, indicating that AR can tolerate substantial SAE imperfections in practice. We will clarify this in the revised version and explicitly position advances in monosemantic SAEs as complementary to AR.
>
> ## W2+Q1 (unclear scalability of AR to high-dimensional activations and long-context models)
> First, AR already operates on high-dimensional SAE spaces, but at inference time we never reason over the full latent dimension. We only track a small dictionary of task-relevant concepts, resulting in matrix $A \in \mathbb{R}^{|D|\times|S|}$, where $|S|$ is the context length and $|D|$ the number of concepts. Thus complexity scales with $|D|$ and $|S|$, not with SAE size; the expensive feature search happens once on the train set.
>
> For long contexts, the main cost is storing and updating $A$. A dense, pessimistic estimate (128k tokens, 1,000 concepts) is ≈0.5GB, which is small compared to the backbone LLM. In practice, $A$ is highly sparse, and the rule set is small, so both memory and compute overhead remain modest.
>
> Importantly, the *discrete* reasoning step is more robust to long contexts than autoregressive text-based reasoning: once concepts are extracted, rule application is independent of token distance. Our multi-hop results (1/3/5 hops Table 1 in paper) already demonstrate that AR remains stable with growing reasoning depth and effective complexity. We will add a short scalability and long-context discussion to clarify this.
>
>
> ## W3 (integration with probabilistic or continuous reasoning paradigms)
> We agree this is an important direction. AR is actually solver-agnostic by design; other logic solvers can be integrated without architectural changes. Thus, probabilistic or continuous engines such as SLIPCOVER [2] or ProbFOIL [3] would allow the reasoning module to propagate graded SAE activations rather and would allow to better cope with the uncertainties of these activations. We will expand the future work section to highlight this as a natural extension.
>
> [1] Measuring and Guiding Monosemanticity (Härle et al., Neurips 2025)
>
> [2] SLIPCOVER (Bellodi & Riguzzi, Theory and Practice of Logic Programming 15.2, 2015)
>
> [3] ProbFOIL (De Raedt et al., IJCAI, 2015)

---

> > ### Author Response · Authors · 2025-11-19
> >
> > ## Q2 (sensitivity of AR performance to SAE choice and layer placement)
> >
> > Prior work shows that later transformer layers encode more semantic, more monosemantic, and more stable features, whereas earlier layers mainly capture local or syntactic patterns and therefore produce noisier and less interpretable concepts [4,5]. For this reason, existing SAE work typically attaches the autoencoder to late layers (often in the last third of the network), and AR follows this convention.
> >
> > Importantly, AR itself is layer-agnostic. To assess sensitivity, we now include a small layer ablation where AR is applied to the adjacent layers before and after the originally selected attachment points. Across these layers, performance remains highly consistent, confirming that AR functions robustly across different positions in the semantic portion of the model. Our ablation with ($\mathcal R_{single}$) and autothresholding shows stable results for the adjacent layers on the R2C dataset.
> >
> >
> > | layer | $\alpha$ | ACC |
> > |-|-|-|
> > | 18 | 0.8  | 93.67% |
> > | 22 | 0.7  | 93.33% |
> >
> >
> > ## Q3 (generalizability of the framework to multimodal settings)
> > We consider our work foundational in nature, demonstrated here with LLMs, but readily extensible to multimodal applications. In general, AR is agnostic to the underlying source of activations; it operates on concept representations regardless of whether they stem from text, images, or other modalities. Recent works have explored how SAEs perform in VLMs [6,7] or text-to-image models [8], demonstrating that concept detection and steering are feasible in these modalities, too. Since AR's modular design only requires activations as input, integrating these approaches should be directly applicable. The rule definitions, as instantiated in our evaluations, can be adapted in a similar fashion, e.g., in a multi-modal safety context, one could detect nudity or violence, classify them as unsafe and steer generation toward a non-nude, non-violent output. We will add a discussion on multimodal extensions in the revised manuscript.
> >
> >
> > [4] The Birth of Knowledge: Emergent Features across Time, Space, and Scale in Large Language Models (Sawmya et al., 2025)
> >
> > [5] Saebench: A comprehensive benchmark for sparse autoencoders in language model interpretability (Karvonen, et al., 2025)
> >
> > [6] Sparse autoencoders learn monosemantic features in vision-language models. (Pach et al., NeurIPS 2025)
> >
> > [7]  Archetypal SAE: Adaptive and Stable Dictionary Learning for Concept Extraction in Large Vision Models (Fel et al., ICML 2025)
> >
> > [8] SAEmnesia: Erasing Concepts in Diffusion Models with Sparse Autoencoders. (Cassano et al., 2025)

---

> ### Author Response · Authors · 2025-11-27
>
> As the discussion phase is drawing to a close, we would like to ask if there are any remaining concerns or clarifications you would like us to address. We would be glad to provide further details if helpful, and appreciate it if you would reconsider the scoring if no concerns remain. Thank you for your time and consideration.

---

> > ### Comment · Reviewer_aA2t · 2025-11-28
> >
> > Thank you for the comprehensive rebuttal and for addressing my earlier comments. The clarifications strengthened my understanding of the method. I will retain my positive score, as I find the work to offer some substantive contributions to the field.

---

### Official Review · Reviewer_WLZc · 2025-10-28

**Soundness:** 3
**Presentation:** 4
**Contribution:** 3
**Rating:** 8
**Confidence:** 3

**Summary:**

This paper introduces ActivationReasoning (AR) a novel framework to embed logical reasoning over the latent representations of an LLM. AR finds latent concept representations, map them into logical propositions and finally apply logical inference. The framework is evaluated over different reasoning tasks, ranging from multi-hop reasoning to context-sensitive safety, with promising results that improve transparency  and reliability of the model.

**Strengths:**

- The paper is well written and clear to follow.
- The introduced framework is very valuable and the problem studied very important for the advancement of Large AI models in general.
- Extensive experimental evaluations.
- Interesting discussion of AR's limitations.

**Weaknesses:**

- The rules need to be known in advance, and this may restrict the model applicability, as this is not often the case. Also several rules in real-world settings are only partially true, and then applying forward chaining may introduce some errors.


MINOR COMMENTS

- "On our novel Rail2Country dataset (..)", I think this dataset could be explicitly mentioned as a new contribution of the paper.
- Fig. 2 is referenced before Fig. 1 at page 2, while it only appears on page 4.
- "Only a few provide a formal mechanism to compose concepts into higher-level abstractions or to disambiguate polysemous
features." Here it is seems some references are missing, or it misses an explicit reference to the ones mentioned above. Moreover, I think some other relevant examples of CBMs using logic rules to connect concepts to higher-level downstream tasks, like e.g. [1,2], can also be commented. This is also connected to the paragraph on systems integrating concept-level representations with reasoning modules for interactions and explainability.
- "Neuro-symbolic AI (...) " I'd mention some more recent work, like e.g. [3]
- "A Relational-feature representation (R relation) defines a concept by a set of relations among SAE features". Could you please explain/extend more how Relational features work? I understand that e.g. the concept "hate" is connected to multiple features, like the feature connected to the concept "slurs" and another feature connected to the concept "demeaning stereotypes". Hence, it's not clear to me how it is concretely different from multi-feature representations.
- "For example, features consistently activating for the notion of a Bridge can be directly assigned." Not clear how.. could you please better explain this? You mean by inspection over all the possible activations? I think in this case it can be unfeasible for large representation spaces.
- While not strictly necessary as the paper is very clear, I think to make the paper more self-consistent a brief section about background, e.g. on SAEs or CBMs could be useful.
- "initialize AR with Rmulti) using" -> right bracket is a typo


[1] Barbiero, Pietro, et al. "Interpretable neural-symbolic concept reasoning." International Conference on Machine Learning. PMLR, 2023.
[2] Barbiero, Pietro, et al. "Relational concept bottleneck models." Advances in Neural Information Processing Systems 37 (2024): 77663-77685.
 [3] Marra, Giuseppe, et al. "From statistical relational to neurosymbolic artificial intelligence: A survey." Artificial Intelligence 328 (2024): 104062.

**Questions:**

1) The list of concepts to build the dictionary should be pre-defined or can be learnt as well? E.g. by using an approach similar to label-free CBMs?
2) The concepts need to be known in advance wether having a single, multiple or relational representation, or it can be learnt automatically? I mean, in the automatic setting the representation is learnt, but you should know if r_c belongs either to R_single, R_multi or R_relation? In this case this may introduce a relevant bias in the learning and need additional knowledge about each concept range.
3) I was wondering if you tried or you think it make sense for a concept c, to have r_c = [r_cs,r_cm,r_cr], namely to allow a concept to have simultaneously a single, multiple and relational representation. In this way probably any concept can manage the mono and poly semantic.
4) If I understand correct A and A' have the same dimension, i.e. the concept inferrable are already pre-fixed (even if not learnt) from the beginning. In this way it seems like a semi-supervised setting where some concepts are supervised and others (the inferred) are not. As the rules are already known, what is the strong difference on just applying forward chaining as a pre-processing step and then use the supervisions derived at training time?
5) How you decide to attach AR after layer 23 and 20 on the two backbone models? Why not, e.g. layer 15?

---

> ### Author Response · Authors · 2025-11-19
>
> We sincerely thank the reviewer for their thoughtful feedback. Below we address the main substantive questions; we will directly incorporated fixes for typos and minor issues into the revised manuscript.
>
> ## W1 (Rules must be known in advance and real-world rules)
> Actually, rules need not be known in advance. While our experiments use rules derived from dataset structure (which allowed us to focus on demonstrating AR's core reasoning capabilities), AR is agnostic to rule origin. Rules can be flexibly obtained through various means, e.g., extracted by prompting LLMs to propose candidate rules from natural language descriptions, derived from domain experts, or learned from data, and then plugged directly into AR without architectural changes. The key distinction is that AR separates rule specification from model training, enabling transparent and auditable reasoning. Thus, unlike end-to-end neural approaches where logic remains entangled and opaque, AR makes rules explicit and modifiable. This is particularly valuable, e.g., in safety-critical domains (e.g., guard models) where requirements evolve. We will discuss this "manual initialization to automatic rule discovery" trajectory more prominently in our revised Future Work section (§6).
>
> We agree incorporating "uncertain rules" is an important consideration. AR's current implementation uses classical forward chaining, which assumes crisp logical truth values. However, the framework naturally extends to probabilistic or fuzzy reasoning by replacing the solver backend. This would enable AR to propagate uncertainties through rule chains and output confidence-weighted conclusions, directly addressing scenarios where rules hold only approximately or contextually. We will position this extension more prominently in our revised Future Work section (§6) as a natural next step toward handling noisy, real-world rule systems.
>
> ## C1 (Rail2Country as contribution)
> Agreed, we will position our contribution regarding the newly introduced dataset Rail2Country more prominently in the revised paper.
>
>
> ## C3+C4 (Additional citations)
> We thank the reviewer for the valuable pointers and will add the above mentioned citations.
>
>
> ## C5 (Differentiation of $\mathcal R_{multi}$ and $\mathcal R_{relation}$)
> The key difference is how features are combined:
> - $\mathcal R_{multi}$: Features contribute independently through weighted aggregation (essentially a linear combination). No feature interactions are modeled.
> - $\mathcal R_{relation}$: Encodes structured interactions via a shallow decision tree, enabling conjunctions (e.g., both slurs and stereotypes), contextual exclusions (e.g., filtering quoted/educational mentions), and feature-dependent thresholds.
>
> Example: For "hate speech," $\mathcal R_{multi}$ simply sums slur and stereotype features, which false-positives on "The article discusses how slurs and stereotypes harm communities." $\mathcal R_{relation}$ can encode: "slurs AND stereotypes, BUT NOT in educational context", explaining its stronger performance in Table 2, especially on context-sensitive categories.
> We will add this clarification to the paper.
>
> ## C6 (Clarification for manual feature to concept assignment)
> "Directly assigned" refers to cases where the feature-concept connection is immediately apparent. For example, by sampling a few sentences where a concept like "Bridge" appears and observing that a specific SAE feature consistently activates strongly across these instances, one can directly assign that feature.
> However, we agree that this approach becomes impractical for large SAE spaces. This is why our framework supports automatic concept construction (see submission Eq. 1): given token-level labels, we identify discriminative SAE features automatically. This data-driven approach scales naturally and was used for several experiments in the paper.
> Manual assignment is thus mentioned for completeness but automatic extraction is the practical default. We will clarify this in the revision.

---

> > ### Author Response · Authors · 2025-11-19
> >
> > ## Q1 (Predefined and Learnable Concept Dictionary)
> > Yes, similar to label-free CBMs the SAE has a lot of unused concepts, we are just using a small portion of them. So if at a later stage we need additional or other concepts we can search for them in the SAE and add them to the concept dictionary. Moreover, several Auto-interp. methods have emerged for SAEs [1] in addition to the ones we propose to find concepts in the SAE latent.
> >
> > ## Q2+Q3 (Combination and automation of Concept Representations)
> > Currently, the representation type (single/multi/relational) is fixed beforehand for each concept. However, adding the capability for automatic representation selection is a valuable future step to increase expressivity. A potential approach would mirror our auto-thresholding: on the training set, select the representation type that achieves the best balanced accuracy for each concept.
> > We agree that hybrid representations are a very interesting idea that could help manage mono- and polysemantic cases. It would require additional mechanisms to determine which representation to activate in a given context.
> > We highlight both directions in our revised future work section.
> >
> > ## Q4 (Pre- vs. Post-computing rule outcomes)
> > This is an important distinction to clarify.
> > The key difference is that AR composes concepts without requiring labels for the final concepts. Pre-computing would need labeled training examples for concepts like "Golden Gate Bridge", AR infers it zero-shot from "Bridge ∧ San Francisco ∧ USA" via rules.
> > Additionally, in our experiments rules change per sample (PrOntoQA, ProverQA), making pre-processing infeasible. Even where rules are stable, AR enables post-deployment modification without retraining.
> >
> > In case, we have misunderstood the reviewer's initial question, we hope they could clarify further.
> >
> > ## Q5 (Choice of SAE attatched Layer)
> > Prior work shows that later transformer layers encode more semantic, more monosemantic, and more stable features, whereas earlier layers mainly capture local or syntactic patterns and therefore produce noisier and less interpretable concepts [2,3]. For this reason, existing SAE work typically attaches the autoencoder to late layers (often in the last third of the network), and AR follows this convention.
> >
> > Importantly, AR itself is layer-agnostic. To assess sensitivity, we now include a small layer ablation where AR is applied to the adjacent layers before and after the originally selected attachment points. Across these layers, performance remains highly consistent, confirming that AR functions robustly across different positions in the semantic portion of the model. Our ablation with ($R_{single}$) and autothresholding shows stable results for the adjacent layers on the R2C dataset.
> >
> >
> > | layer | $\alpha$ | ACC |
> > |-|-|-|
> > | 18 | 0.8 | 93.67% |
> > | 22 | 0.7 | 93.33% |
> >
> >
> > [1] Automatically Interpreting Millions of Features in Large Language Models (Paulo et al., ICML 2025)
> >
> > [2] The Birth of Knowledge: Emergent Features across Time, Space, and Scale in Large Language Models (Sawmya et al., 2025)
> >
> > [3] Saebench: A comprehensive benchmark for sparse autoencoders in language model interpretability (Karvonen, et al., 2025)

---

> > > ### Comment · Reviewer_WLZc · 2025-11-25
> > >
> > > I thanks a lot the authors for the clarifications and the satisfactory answers to my questions. I have also read the other reviews and the rebuttal. Overall I think the pro of this paper are substantially higher than cons and thus I'd like to keep my acceptance score.

---

> > > > ### Author Response · Authors · 2025-11-25
> > > >
> > > > We appreciate your efforts into the encouraging review and the overall positive score!

---

### Official Review · Reviewer_vdgu · 2025-10-31

**Soundness:** 2
**Presentation:** 2
**Contribution:** 2
**Rating:** 2
**Confidence:** 4

**Summary:**

This paper proposes ActivationReasoning (AR), a framework that claims to perform logical reasoning directly from the latent activation space of LLMs. It contains 3 main steps: (1) use SAE to extract “concept”; (2) map the activations to propositional symbol via thresholding; (3) apply user-defined rules over the symbols.

**Strengths:**

The paper presents a clear pipeline and demonstrate potential applications (like safety). The visualisation of their method is clear. It could inspire how to construct reliable symbolic reasoning using latent representations.

**Weaknesses:**

1. Concept extraction is loosely defined and not validated.

2. Hard-coded rules, dataset-based (using training set) concept extraction from SAE output.

3. Lack ablation on concept accuracy / why baseline model failed, is it because   of lacking the knowledge from the rules, or it is reasoning capability issue. Ablation is needed to understand why such huge performance gap of ~50\% occurs.

**Questions:**

Are the logical rules fully hand-written, and how are they chosen or validated for each dataset?

---

> ### Author Response · Authors · 2025-11-19
>
> We thank reviewer vdgu for taking the time to review our manuscript.
> We note that several points appear to stem from misunderstandings, which we address in detail below.
>
> ## W1 (concept extraction is loosely defined and not validated)
> Concept extraction is indeed an important part of Activation Reasoning and is **explicitly defined and explained** in **Section 3.1** with a distinguished paragraph, **“Constructing Representations”**. There we **define** how each concept's latent representation is built via $\mathcal R_{\text{single}}, \mathcal R_{\text{multi}}$, or $\mathcal R_{\text{relation}}$, how features are selected on the train set, and how thresholds are chosen. Furthermore, the quality of concept detection is **explicitly validated**: **Table 5** reports concept-level accuracy, and **Table 2** shows how different representation types systematically affect performance. Additional details per experimental evaluation are denoted in the corresponding subsections of App A. We further note that other reviewers (WLZc, aA2t) specifically commented that the pipeline and its components are clearly described.
> If there are specific steps of the concept extraction or validation procedure that remain unclear, we would be grateful if the reviewer could point them out so that we can address them explicitly in a revised version.
>
> ## W2+Q1 (hard-coded rules and dataset-dependent concept extraction from SAE outputs)
> Actually, AR does not rely on "hard-coded" rules. Rules are flexible and can be obtained via various means (e.g., LLM translation, domain experts) and updated at inference time without retraining the LLM or SAE. In our reasoning experiments (PrOntoQA, ProverQA), the rule set dynamically changes with each sample, AR simply executes whichever rules the instance provides.
> For structured benchmarks, rule extraction is straightforward (often directly provided in the dataset format). However, AR naturally extends to automatic rule generation: one can prompt LLMs to propose candidate rules and plug them directly into AR without architectural changes. While we focus on demonstrating AR's core reasoning capabilities in this work, we do explicitly discuss this "manual initialization → automatic rule discovery" trajectory in our Limitations (Sec. 5) and Future Work section (Sec. 6).
> The key distinction is that AR separates rule specification from model training, enabling transparent, auditable, and modular reasoning that can be adapted post-deployment, unlike end-to-end neural approaches where such logic remains entangled and opaque.
>
> ## W3 (missing ablation to explain performance gap and diagnose baseline failure (knowledge vs. reasoning))
>
> We do **explain and diagnose** the **baseline failure** in **Table 5**. Here, we evaluate concept accuracy, showing that the extracted concepts are accurate on held-out data. Moreover, at inference time both models (AR and Base) do in fact receive the same input information (same background, rule and task information). Thus, the substantial performance difference which we observe cannot be attributed to different input, but is due to the LLM’s  reasoning limitations in comparison to explicit reasoning with AR.
> Nevertheless, we have run an additional ablation in which the LLM receives the rules in exactly the same logical notation as for AR, (see Table below: Additional Baselines (Llama3.1 8B and Gemma3-8B **w/ it (normed)**)). Across all datasets we observe a modest increase in accuracy, however still remaining far below the AR runs indicating that the representation format itself is not the primary bottleneck of the base LLMs. Rather, the limitation appears to stem from the reasoning process, not from rule presentation.
>
> ### Additional Baselines
> | mode | prontoqa 1-hop ACC | prontoqa 3-hop ACC | prontoqa 5-hop ACC | r2c mono ACC | r2c meta ACC | proverqa easy ACC | proverqa medium ACC | proverqa hard ACC |
> | - | - | - | - | - | - | - | - | - |
> | **Llama3.1 8B**  | 51.0% | 50.8% | 50.3% | 41.0% | 29.7% | 43.6% | 33.6% | 36.8% |
> | w/ it (normed)   | 49.3% | 48.0% | 50.6% | 47.0% | 30.3% | 49.6% | 53.6% | 47.6% |
>
> | mode | prontoqa 1-hop ACC | prontoqa 3-hop ACC | prontoqa 5-hop ACC | r2c mono ACC | r2c meta ACC | proverqa easy ACC | proverqa medium ACC | proverqa hard ACC |
> | - | - | - | - | - | - | - | - | - |
> | **Gemma2 9B**    | 48.5% | 47.5% | 47.9% | 35.3% | 25.7% | 39.4% | 29.8% | 25.8% |
> | w/ it (normed)   | 48.6% | 48.0% | 48.1% | 61.7% | 54.7% | 35.8% | 41.6% | 34.0% |

---

> > ### Comment · Reviewer_vdgu · 2025-11-27
> >
> > Thanks for the clarification. Formally defining a concept and explaining how to obtain it are different tasks. Overall, there is still room for the paper to improve its presentation and explain these matters more clearly

---

> ### Author Response · Authors · 2025-11-27
>
> We thank the reviewer for their time and constructive feedback, and we agree that explicitly defining a concept improves the clarity of the paper. To address this, we have revised Section 3.1 and now include a formal definition immediately before describing the instantiation of representations. We follow the general spirit of concept-based models, where concepts are encoded in latent spaces [1,2,3,4] and provide an explicit formalization for ActivationReasoning:
>
> > **Formalizing Concepts.** We formally define a concept $c$ as a tuple $(n_c,r_c,\tau_c)$. Let $\mathcal{L}\subseteq\mathbb{R}^d$ denote the latent space of concept codes produced by a fixed encoder $E$ applied to the model hidden states. For each token $t$ with hidden state $h_t$, we obtain a latent code via the encoder $\ell_t=E(h_t)\in\mathcal{L}$. Concretely: (i) $n_c$ is a semantic identifier for the concept (e.g., "Bridge"). (ii) $\tau_c\in[0,\infty)$ is a soft activation threshold that determines when the concept is considered active. (iii) $r_c:\mathcal{L}\rightarrow[0,\infty)$ is the concept latent representation in this space. Implemented as a function, it maps latent codes $\ell_t\in\mathcal{L}$ to a non-negative activation score $a(c,t):=r_c(\ell_t)\in[0,\infty)$. The collection of all concepts used by AR forms the concept dictionary $\mathcal{D}=\{c_i\}_{i=1}^{N}$.
>
> We believe that our rebuttal has fully addressed the concerns raised. If there are any additional concrete points that require clarification or revision, we would be very happy to address them. In the absence of further specific concerns, we kindly ask the reviewer to reconsider the score in light of the rebuttal and the revisions.
>
>
> [1] Promises and Pitfalls of Black-Box Concept Learning Models, Mahinpei et. al. (ICML 2021)
>
> [2] Concept Whitening for Interpretable Image Recognition, Chen et. al. (Nature 2020)
>
> [3] Transformer Feed-Forward Layers Build Predictions by Promoting Concepts in the Vocabulary Space, Geva et. al. (ACL 2022)
>
> [4] The Value of Symbolic Concepts for AI Explanations and Interactions, Stammer et. al.

---

### Author Response · Authors · 2025-11-19
**Thanks to the reviewers!**

We thank all reviewers for their valuable time and effort, and provide individual responses below. We further wish to note that we will update the submission in the following days according to our responses.

---

> ### Author Response · Authors · 2025-11-25
> **Manuscript updated**
>
> Dear reviewers,
>
> We have uploaded the revised version of our manuscript. All changes are highlighted in blue. In particular, we have:
>
> - Added ablation studies on selected hyperparameters and SAE placement
> - Included additional baseline comparisons
> - Expanded the discussion of limitations and future work
> - Improved the wording in the methods section for greater clarity
>
> We appreciate the reviewers' helpful feedback and careful reviews.

---

### Author Response · Authors · 2025-12-03
**Rebuttal Summary**

We would like to provide a brief update on the review process.

Overall, reviewers emphasized that our work “is very valuable and important for the advancement of Large AI models” (WLZc). Key strengths highlighted include strong multi-hop robustness, logic-based safety composition, and low runtime overhead (tgpL). Several reviewers also noted that the paper is well written, with a clear pipeline and method visualization (vdgu, WLZc, aA2t).

We believe reviewer vdgu’s rejection stems from misunderstandings and overlooking. Validation of Concept extraction and failure analysis (W1 and W3) can be found in the appendix (Section F and Table 5) and lines 406-408. Regarding W2+Q1, we clarify that rules are not hard-coded but are dynamic and updateable at inference time without retraining (lines 494-496).

In response to the feedback, we added hyperparameter/SAE ablations (tgpL, WLZc, aA2t), and prompting baselines (tgpL), clarified dataset setup (tgpL), and incorporated suggested future directions, e.g., multimodal settings (aA2t), probabilistic reasoning (WLZc), and inductive/abductive reasoning (tgpL). A summary of where concerns are addressed is provided below. We believe all concerns raised have now been resolved, and the revision has significantly strengthened the paper. We are grateful to the reviewers for their constructive feedback.

Due to the early closure of the review period, Reviewers vdgu and tgpL could not acknowledge the full rebuttal. However, the other reviewers responded positively, praising the “comprehensive rebuttal” (aA2t). Reviewer WLZc explicitly noted having read all reviews before concluding that the “pros of this paper substantially outweigh the cons,” confirming that the critiques do not diminish the paper’s core contribution.

| Addressed Concerns| Location in revised paper (section/lines) |
| -------------------------------------------------------- | ------------------------------------------- |
| SAE layer ablation |Section G |
| Additional LLM sampling comparisons | Section E and Table 4 |
| Hyperparameter ablations | Section H and Table 6|
| SAE-induced limitations | lines 480-485 and 505-508|
| Scalability (high-dim, long-context) | lines 473-476|
| Formalized concept definition | lines 186-194|
| Train–test protocol | lines 893-909|
| Probabilistic reasoning, induction/abduction reasoning, multimodal extensions | lines 494-504, 524-531|

---

### Meta-Review · Area_Chair_vFKJ · 2026-01-11

**Summary:**

The reviewers’ concerns were around:
1. Presentation / Formal definitions: definition/validation of concept extraction (vdgu), integration e.g. with probabilistic programming (aA2t)
2. Issues with rules: hard-coded rules (vdgu), need for them to be known (WLZc), (tgpL)
3. Foundational about the method: inheriting limitations of SAE (aA2t), performance in high-dimensional space or with long contexts (aA2t), relation of AR to “True” reasoning models (tgpL)
4. Experiments: Requesting more ablations (vdgu), “double dipping” (tgpL).

**Reviewer Concerns:**

The rebuttal was extensive and helped with additional ablations; explanations; as well as explicitly acknowledging limitations in a faithful way.

Specifically:
1. Presentation / Formal definitions: The concerns have been partially addressed, for example by adding a formal definition for concept extraction as well as validation.
2. Issues with rules: It seems that there was a mix of misunderstandings or limitations which were not clearly acknowledged. This seems to mostly resolve the concerns, and both reviewers vdgu and WLZc either explicitly say they are happy with the authors' reply or they do not mention in their follow-up any remaining issues with rules.
3. Foundational about the method: These are good points but really to a large extent it is about the scope of the paper. The authors provide convincing explanations, for example for the case of deduction vs abduction, the authors explain how the reasoning component could be replaced with abductive engines. The authors also acknowledge certain limitations. Overall, I see a combination of: acknowledging limitations, putting the question in context of the current scope of the paper and related work, and explaining how it could be extended in the future; this helps us accept that to a large extent, some of these concerns can indeed be regarded as future work.

**Reviewer Scores:**

vdgu: the reviewer replies mentioning that there is still room to improve the paper. However, I do not see a strong argument about how the rebuttal has failed to completely move the reviewer move from their very low score of “2” (especially since after the reviewer’s follow-up, the authors reply again with more evidence). I would expect that a full discussion period would allow for reviewer vdgu to significantly raise their score to at least borderline.

WLZc: The reviewer already gives a high score and seems happy with the rebuttal.

aA2t: The reviewer replies to the rebuttal acknowledging that it has helped. The reviewer does not indicate an explicit raising in their score, neither do they indicate any remaining concerns. I expect that a full discussion would have nudged this reviewer to further raise their score.

tgpL: The reviewer has not answered to the rebuttal. I think some concerns have been addressed while some have remained unresolved. At the same time, most of the unresolved issues regard topics that the authors have, in my view, successfully argued that they constitute reasonable future work. Therefore, I expect this reviewer to have increased their score given more discussion.

---

### Decision · Program_Chairs · 2026-01-26

Accept (Poster)